# FedMix: Approximation of Mixup under Mean Augmented Federated Learning

**Tehrim Yoon & Sumin Shin & Sung Ju Hwang & Eunho Yang**
Korea Advanced Institute of Science and Technology (KAIST)
Daejeon, South Korea
`{tryoon93,sym807,sjhwang82,eunhoy}@kaist.ac.kr`

## Abstract

Federated learning (FL) allows edge devices to collectively learn a model without directly sharing data within each device, thus preserving privacy and eliminating the need to store data globally. While there are promising results under the assumption of independent and identically distributed (iid) local data, current state-of-the-art algorithms suffer from performance degradation as the heterogeneity of local data across clients increases. To resolve this issue, we propose a *simple* framework, *Mean Augmented Federated Learning (MAFL)*, where clients send and receive *averaged* local data, subject to the privacy requirements of target applications. Under our framework, we propose a new augmentation algorithm, named *FedMix*, which is inspired by a phenomenal yet simple data augmentation method, Mixup, but does not require local raw data to be directly shared among devices. Our method shows greatly improved performance in the standard benchmark datasets of FL, under highly non-iid federated settings, compared to conventional algorithms.

## 1 Introduction

As we enter the era of edge computing, more data is being collected directly from edge devices such as mobile phones, vehicles, facilities, and so on. By decoupling the ability to learn from the delicate process of merging sensitive personal data, Federated learning (FL) proposes a paradigm that allows a global neural network to learn to be trained collaboratively from individual clients without directly accessing the local data of other clients, thus preserving the privacy of each client (Konečný et al., 2016; McMahan et al., 2017). Federated learning lets clients do most of the computation using its local data, with the global server only aggregating and updating the model parameters based on those sent by clients.

One of the standard and most widely used algorithm for federated learning is FedAvg (McMahan et al., 2017), which simply averages model parameters trained by each client in an element-wise manner, weighted proportionately by the size of data used by clients. FedProx (Li et al., 2020b) is a variant of FedAvg that adds a proximal term to the objective function of clients, improving statistical stability of the training process. While several other methods have been proposed until recently (Mohri et al., 2019; Yurochkin et al., 2019; Wang et al., 2020), they all build on the idea that updated model parameters from clients are averaged in certain manners.

Although conceptually it provides an ideal learning environment for edge devices, the federated learning still has some practical challenges that prevent the widespread application of it (Li et al., 2020a; Kairouz et al., 2019). Among such challenges, the one that we are interested in this paper is the heterogeneity of the data, as data is distributed non-iid across clients in many real-world settings; in other words, each local client data is not fairly drawn from identical underlying distribution. Since each client will learn from different data distributions, it becomes harder for the model to be trained efficiently, as reported in (McMahan et al., 2017). While theoretical evidence on the convergence of FedAvg with non-iid case has recently been shown in (Li et al., 2020c), efficient algorithms suitable for this setting have not yet been developed or systematically examined despite some efforts (Zhao et al., 2018; Hsieh et al., 2020).

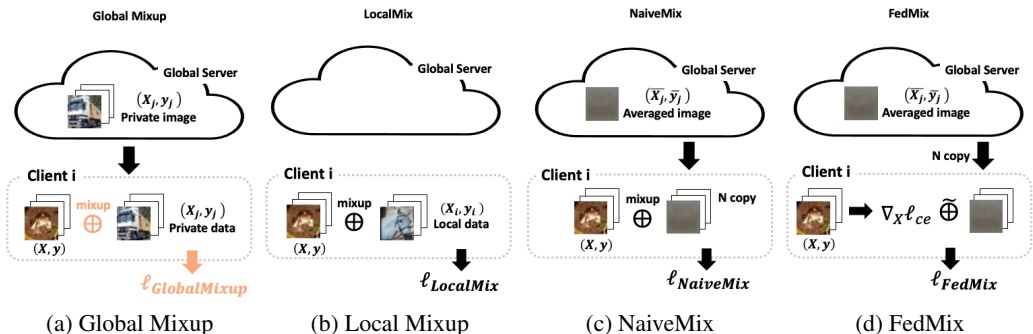

(a) Global Mixup      (b) Local Mixup      (c) NaiveMix      (d) FedMix

Figure 1: Brief comparisons of Mixup strategies in FL and MAFL. (a) Global Mixup: Raw data is exchanged and directly used for Mixup between local and received data, which violates privacy. (b) Local Mixup: Mixup is only applied within client's local data. (c) NaiveMix: Under MAFL, Mixup is performed between local data and received *averaged* data. (d) FedMix: Under MAFL, our novel algorithm approximates Global Mixup using input derivatives and *averaged* data.

In addition to non-iid problem, another important issue is that updating model parameters individually trained by each client is very costly and becomes even heavier as the model complexity increases. Some existing works Smith et al. (2016); Sattler et al. (2019) target this issue to decrease the amount of communications while maintaining the performance of FedAvg. A more practical approach to reduce communication cost is to selectively update individual models at each round, rather than having all clients participate in parameter updates. This partial participation of clients per round hardly affects test performance in ideal iid settings but it can exacerbate the heterogeneity of weight updates across clients and as a result, the issue of non-iid (McMahan et al., 2017).

In order to mitigate the heterogeneity across clients while protecting privacy, we provide a novel yet simple framework, mean augmented federated learning (MAFL), in which each client exchanges the updated model parameters as well as its *mashed* (or averaged) data. MAFL framework allows the trade-off between the amount of meaningful information exchanged and the privacy across clients, depending on several factors such as the number of data instances used in computing the average. We first introduce a naive approach in our framework that simply applies Mixup (Zhang et al., 2018) between local data and averaged external data from other clients to reduce a myopic bias.

Here, we go further in our framework and ask the following seemingly impossible question: can *only averaged data* in our framework that has lost most of the discriminative information, bring the similar effect as a *global Mixup* in which clients directly access others' private data without considering privacy issues? Toward this, we introduce our second and more important approach in our framework, termed Federated Mixup (FedMix), that simply approximates the loss function of global Mixup via Taylor expansion (it turns out that such approximation only involves the averaged data from other clients!). Figure 1 briefly describes the concept of our methods.

We validate our method on standard benchmark datasets for federated learning, and show its effectiveness against the standard federated learning methods especially for non-iid settings. In particular, we claim that FedMix shows better performance and smaller drop in accuracy with more heterogeneity or fewer clients update per communication round, further increasing difficulty of federated learning.

Our contribution is threefold:

- We propose a simple framework for federated learning that averages and exchanges each local data. Even naive approach in this framework performing Mixup with other clients' *mashed* data shows performance improvement over existing baselines on several settings.

- We further develop a novel approximation for insecure global Mixup accessing other clients' local data, and find out that Taylor expansion of global Mixup only involves the averaged data from other clients. Based on this observation, we propose FedMix in our framework approximating global Mixup without accessing others' raw data.

- We validate **FedMix** on several FL benchmark datasets especially focusing on non-iid data settings where our method significantly outperforms existing baselines while still preserving privacy with minimal increases in communication cost.

## 2    RELATED WORK

**Federated learning**   Federated learning was first proposed in Konečný et al. (2016) where the prevalent asynchronous SGD (Dean et al., 2012) is used to update a global model in a distributed fashion. A pioneering work in this field proposed the currently most widely used algorithm, FedAvg (McMahan et al., 2017), which is also the first synchronous algorithm dedicated to federated setting. Shortly after, Li et al. (2020b) proposed a variant of FedAvg, named FedProx, where the authors claimed to overcome statistical heterogeneity and increase stability in federated learning. Recent studies attempt to expand federated learning with the aim of providing learning in more diverse and practical environments such as multi-task learning (Smith et al., 2017), generative models (Augenstein et al., 2020), continual learning (Yoon et al., 2020), semi-supervised learning (Jeong et al., 2020), and data with noisy labels (Tuor et al., 2020). Our paper focuses on general federated settings, but it could be considered in such various situations.

However, these algorithms may obtain suboptimal performance when clients participating in FL have non-iid (Zhao et al., 2018; Hsieh et al., 2020) distributions. While the convergence of FedAvg on such settings was initially shown by experiments in McMahan et al. (2017) and later proved in Li et al. (2020c), it does not guarantee performance as good as it would have been for iid setting. Existing algorithms that pointed out this issue have major limitations, such as privacy violation by partial global sharing of local data (Zhao et al., 2018) or no indication of improvement over baseline algorithms such as FedAvg (Hsieh et al., 2020). Our method aims to improve performance particularly on these non-iid situations, without compromising privacy.

**Mixup**   Mixup (Zhang et al., 2018) is a popular data augmentation technique that generates additional data by linear interpolation between actual data instances. Mixup has been usually applied to image classification tasks and shown to improve test accuracy on various datasets such as CIFAR10 and ImageNet-2012 (Russakovsky et al., 2015), and, on popular architectures such as ResNet (He et al., 2016) and ResNeXt (Xie et al., 2017), for various model complexity. It is also reported in Zhang et al. (2018) that Mixup helps with stability, adversarial robustness (Zhang et al., 2018), calibration, and predictive certainty (Thulasidasan et al., 2019). Mixup is expanding from various angles due to its simplicity and popularity. First, beyond image classification tasks, its effectiveness has been proven in various domains such as image segmentation (Eaton-Rosen et al., 2020), speech recognition (Warden, 2018), and natural language processing (Guo et al., 2019). Also, several extensions such as Manifold Mixup (Verma et al., 2018), which performs Mixup in latent space, or CutMix (Yun et al., 2019), which replaces specific regions with others patches, have been proposed.

In most of the previous studies on federated learning, Mixup was partially (or locally) used as a general data augmentation technique. Some recent studies (Oh et al., 2020; Shin et al., 2020) proposed to send blended data to server using Mixup, but they require sending locally- and linearly-mixed (mostly from two instances) data to server at every round, therefore being susceptible to privacy issues with huge communication costs. Our work properly modifies Mixup under the restrictions of federated learning and mitigates the major challenges of federated learning such as non-iid clients.

## 3    MEAN AUGMENTED FEDERATED LEARNING (MAFL) AND FEDMIX

We now provide our framework exchanging averaged data for federated learning and main method approximating insecure global Mixup under our framework, after briefly introducing the setup.

### 3.1    SETUP AND BACKGROUND

**Federated learning and FedAvg**   Federated Averaging (FedAvg) (McMahan et al., 2017) has been the most popular algorithmic framework for federated learning. For every communication round $t = 0, \ldots, T - 1$, a client $k \in 1, \ldots, N$ selected for local training sends back its model $\boldsymbol{w}_t^k$ (or only difference to reduce communication cost) to a global server. For every round, $K$ number of clients are selected to locally update and send model parameters. The server simply averages parameters received, so that the global model $\boldsymbol{w}_t$ after $t$ rounds of communications becomes $\boldsymbol{w}_t = \sum_{k=1}^{n} p_k \boldsymbol{w}_t^k$ where $p_k$ is the importance of client $k$ based on the relative number of data in $k$ among all selected clients at $t$. The updated global model is sent back to clients for the next round, which undergoes the following $E$ local updates via stochastic gradient descent (SGD):

$\boldsymbol{w}_{t+1,i+1}^k \leftarrow \boldsymbol{w}_{t+1,i}^k - \eta_{t+1}\nabla\ell(f(\boldsymbol{x}_i^k;\boldsymbol{w}_{t+1,i}),y_i^k)$ for $i = 0, 1, \ldots, E-1$, batch size $B$, and local learning rate $\eta$. Here, $\ell$ is the loss function for learning and $f(\boldsymbol{x};\boldsymbol{w}_t)$ is the model output for input $\boldsymbol{x}$ given model weight $\boldsymbol{w}_t$.

**Mixup**   Mixup (Zhang et al., 2018) is a simple data augmentation technique using a linear interpolation between two input-label pairs $(\boldsymbol{x}_i, y_i)$ and $(\boldsymbol{x}_j, y_j)$ to augment $\tilde{\boldsymbol{x}} = \lambda\boldsymbol{x}_i + (1-\lambda)\boldsymbol{x}_j$ and $\tilde{y} = \lambda y_i + (1-\lambda)y_j$. The variable $\lambda \in [0,1]$ is a hyperparameter that is chosen from the beta distribution for each training step.

## 3.2   MAFL: MEAN AUGMENTED FEDERATED LEARNING

The most obvious and powerful way for local models to receive information about data from other clients is to simply receive raw individual data. However, under typical federated setting, each client does not have direct access to individual external data due to privacy constraints, leading to overall performance degradation. We propose a federated learning framework that relaxes the limitation of accessing others' raw data and allows a more granular level of privacy depending on applications. In our new framework, termed mean augmented federated learning (MAFL), clients not only exchange model parameters but also its *mashed* (or averaged) data.

In MAFL, only the part that exchanges averaged data of each client has been added to the standard FL paradigm (Algorithm 1). Here, the number of data instances used in computing the average, $M_k$, controls the key features of MAFL such as privacy and communication costs. Lower $M_k$ value results in more relevant information passed over, but only in cost of less secure privacy and larger communication cost. In one extreme of $M_k = 1$, raw data is thoroughly exchanged and privacy is not protected at all, which is clearly inappropriate for FL. But, in the other extreme, all data of each client is averaged to ensure a considerable degree of privacy. In addition, it also has an advantage on communication cost; each client sends a set of $n_k/M_k$ averaged data where $n_k$ is local data size of client $k$. The remaining question is whether it is possible to improve performance even when exchanging information that is averaged from all local data and loses discriminative characteristics.

The most naive way we can consider in our MAFL framework is to directly use the *mashed* data from other clients, just like regular local data. However, since mashed data has a lot less usable

---

**Algorithm 1:** Mean Augmented Federated Learning (MAFL)

**Input:** $\mathbb{D}_k = \{\boldsymbol{X}_k, \boldsymbol{Y}_k\}$   for   $k = 1, \ldots, N$
$M_k$: number of data instances used for computing average $\bar{\boldsymbol{x}}, \bar{\boldsymbol{y}}$

Initialize $w_0$ for global server
**for** $t = 0, \ldots, T-1$ **do**
  **for** *client $k$ with updated local data* **do**
    Split local data into $M_k$ sized batches
    Compute $\bar{\boldsymbol{x}}, \bar{\boldsymbol{y}}$ for each batch
    Send all $\bar{\boldsymbol{x}}, \bar{\boldsymbol{y}}$ to server
  **end**
  $\mathbb{S}_t \leftarrow K$ clients selected at random
  Send $\boldsymbol{w}_t$ to clients $k \in \mathbb{S}_t$
  **if** *updated* **then**
    Aggregate all $\bar{\boldsymbol{x}}, \bar{\boldsymbol{y}}$ to $\boldsymbol{X}_g, \boldsymbol{Y}_g$
    Send $\boldsymbol{X}_g, \boldsymbol{Y}_g$ to clients $k \in \mathbb{S}_t$
  **end**
  **for** $k \in \mathbb{S}_t$ **do**
    $\boldsymbol{w}_{t+1}^k \leftarrow LocalUpdate(k, \boldsymbol{w}_t; \boldsymbol{X}_g, \boldsymbol{Y}_g)$
  **end**
  $\boldsymbol{w}_{t+1} \leftarrow \frac{1}{K}\sum_{k\in\mathbb{S}_t} p_k\boldsymbol{w}_{t+1}^k$
**end**

---

**Algorithm 2:** FedMix

$LocalUpdate(k, \boldsymbol{w}_t; \boldsymbol{X}_g, \boldsymbol{Y}_g)$ under MAFL (Algorithm 1):
$\boldsymbol{w} \leftarrow \boldsymbol{w}_t$
**for** $e = 0, \ldots, E-1$ **do**
  Split $\mathbb{D}_k$ into batches of size $B$
  **for** *batch*$(\boldsymbol{X}, \boldsymbol{Y})$ **do**
    Select an entry $\boldsymbol{x}_g, \boldsymbol{y}_g$ from
    $\boldsymbol{X}_g, \boldsymbol{Y}_g$
    $\ell_1 =$
    $(1-\lambda)\ell\big(f((1-\lambda)\boldsymbol{X};\boldsymbol{w}), \boldsymbol{Y}\big)$
    $\ell_2 = \lambda\ell\big(f((1-\lambda)\boldsymbol{X};\boldsymbol{w}), \boldsymbol{y}_g\big)$
    $\ell_3 = \lambda\frac{\partial\ell_1}{\partial\boldsymbol{x}}\cdot\boldsymbol{x}_g$
    (derivative calculated at
    $\boldsymbol{x} = (1-\lambda)\boldsymbol{x}_i$ and $y = y_i$ for
    each of $\boldsymbol{x}_i, y_i$ in $\boldsymbol{X}, \boldsymbol{Y}$)
    $\ell = \ell_1 + \ell_2 + \ell_3$
    $\boldsymbol{w} \leftarrow \boldsymbol{w} - \eta_{t+1}\nabla\ell$
  **end**
**end**
**return** $w$

---

information than local data, we can think of a method of mixing it with local data:

$$\ell_{\texttt{NaiveMix}} = (1-\lambda)\ell\Big(f\big((1-\lambda)\boldsymbol{x}_i + \lambda\bar{\boldsymbol{x}}_j\big), y_i\Big) + \lambda\ell\Big(f\big((1-\lambda)\boldsymbol{x}_i + \lambda\bar{\boldsymbol{x}}_j\big), \bar{y}_j\Big) \quad (1)$$

where $(\boldsymbol{x}_i, y_i)$ is an entry from local data and $(\bar{\boldsymbol{x}}_j, \bar{y}_j)$ corresponds to means of (inputs,labels) from other client $j$. Note that Eq. (1) can be understood as the generalization of the loss of directly using the mashed data mentioned above in the sense that such loss can be achieved if $\lambda$ in Eq. (1) is set deterministically to 0 and 1.

In the experimental section, we confirm the effectiveness of MAFL using $\ell_{\texttt{NaiveMix}}$. However, in the next subsection, we will show how to achieve better performance by approximating the global Mixup in a more systematical way in our MAFL framework.

### 3.3    FEDMIX: APPROXIMATING GLOBAL MIXUP VIA INPUT DERIVATIVE

We now provide our main approach in the MAFL framework that aims to approximate the effect of global Mixup only using averaged data from other clients. Consider some client $i$ with its local data $(\boldsymbol{x}_i, y_i)$. It is not allowed in federated learning, but let us assume that client $i$ has access to client $j$'s local data $(\boldsymbol{x}_j, y_j)$. Then, client $i$ would leverage $(\boldsymbol{x}_j, y_j)$ to improve the performance of its local model especially in non-iid settings by augmenting additional data via Mixup:

$$\tilde{\boldsymbol{x}} = (1-\lambda)\boldsymbol{x}_i + \lambda\boldsymbol{x}_j \quad \text{and} \quad \tilde{y} = (1-\lambda)y_i + \lambda y_j. \quad (2)$$

If Mixup rate $\lambda$ is 1, $(\boldsymbol{x}_j, y_j)$ from client $j$ is again directly used like a regular local data, and it would be much more efficient than indirect update of local models through the server.

The essence of our method is to approximate the loss function $\ell\big(f(\tilde{\boldsymbol{x}}), \tilde{y}\big)$ for the augmented data from Eq. (2), with Taylor expansion for the first argument $\boldsymbol{x}$. Specifically, we derive the following proposition:

**Proposition 1** *Consider the loss function of the global Mixup modulo the privacy issues,*

$$\ell_{\texttt{GlobalMixup}}\big(f(\tilde{\boldsymbol{x}}), \tilde{y}\big) = \ell\Big(f\big((1-\lambda)\boldsymbol{x}_i + \lambda\boldsymbol{x}_j\big), (1-\lambda)y_i + \lambda y_j\Big) \quad (3)$$

*for cross-entropy loss $\ell$[1]. Suppose that Eq. (3) is approximated by applying Taylor series around the place where $\lambda \ll 1$. Then, if we ignore the second order term (i.e., $\mathcal{O}(\lambda^2)$), we obtain the following approximated loss:*

$$(1-\lambda)\ell\Big(f\big((1-\lambda)\boldsymbol{x}_i\big), y_i\Big) + \lambda\ell\Big(f\big((1-\lambda)\boldsymbol{x}_i\big), y_j\Big) + \lambda\frac{\partial\ell}{\partial\boldsymbol{x}} \cdot \boldsymbol{x}_j \quad (4)$$

*where the derivative $\frac{\partial\ell}{\partial\boldsymbol{x}}$ is evaluated at $\boldsymbol{x} = (1-\lambda)\boldsymbol{x}_i$ and $y = y_i$.*

While Eq. (4) still involves $\boldsymbol{x}_j$ and $y_j$, invading the privacy of client $j$, the core value of Proposition 1 gets clearer when mixing up multiple data instances from other clients. Note that the vanilla Mixup is not mixing one specific instance with other data, but performing augmentations among several random selected data. In a non-iid FL environment, we can also expect that the effect will be greater as we create Mixup data by accessing as much private data as possible from other clients. From this point of view, let us assume that client $i$ has received a set of $M$ private instances, $J$, from client $j$. Then, the global Mixup loss in Eq. (3) is

$$\frac{1}{|J|}\sum_{j \in J}\ell\Big(f\big((1-\lambda)\boldsymbol{x}_i + \lambda\boldsymbol{x}_j\big), (1-\lambda)y_i + \lambda y_j\Big),$$

and the approximated FedMix loss in Proposition 1 becomes

$$\ell_{\texttt{FedMix}} = \frac{1}{|J|}\sum_{j \in J}(1-\lambda)\ell\Big(f\big((1-\lambda)\boldsymbol{x}_i\big), y_i\Big) + \lambda\ell\Big(f\big((1-\lambda)\boldsymbol{x}_i\big), y_j\Big) + \lambda\frac{\partial\ell}{\partial\boldsymbol{x}} \cdot \boldsymbol{x}_j$$

$$= (1-\lambda)\ell\Big(f\big((1-\lambda)\boldsymbol{x}_i\big), y_i\Big) + \lambda\ell\Big(f\big((1-\lambda)\boldsymbol{x}_i\big), \bar{y}_j\Big) + \lambda\frac{\partial\ell}{\partial\boldsymbol{x}} \cdot \bar{\boldsymbol{x}}_j \quad (5)$$

where we utilize the linearity of Equation 4 in terms of $\boldsymbol{x}_j$ and $y_j$, and $\bar{\boldsymbol{x}}_j$ and $\bar{y}_j$ correspond to mean of $M$ inputs and labels in $J$, respectively. The algorithmic details are provided in the appendix due to the space constraint (see Algorithm 2 in Appendix A).

---

[1]Throughout the paper, we implicitly assume the classification tasks. For regression tasks, we can consider the squared loss function, and the proposition still holds.

### 3.4 PRIVACY ISSUES AND ADDITIONAL COSTS OF MAFL

**Privacy issues of MAFL**    MAFL requires exchanging averaged data by construction. Even though MAFL exchanges only the limited information allowed by the application, it may causes new types of privacy issues. The potential privacy risk of FL or MAFL is beyond the main scope of our study, but in this section, we briefly discuss some basic privacy issues of MAFL and potential solutions.

- There is possibility that local data distribution can be inferred relatively easily from averaged data. This issue simply arises as $M_k$ is not large enough, so that individual data could be inferred from the averaged data easily. On the other hand, if $n_k$ is not big enough, each entry in $X_g, Y_g$ could reveal too much about the whole local distribution of the client it came from.

- It could be easy to infer ownership of each entry in $X_g, Y_g$, if it contains client-id specific information. If clients could identify what other client each entry came from, information about local data of that client could be inferred.

- Additional concerns involve identification of data by detecting change in exchanged averaged data, in case of continual learning, which involves local data change across time. This issue is exacerbated as there is update of averaged data for every minute change on local data, which makes the client receiving $X_g, Y_g$ easier to infer the changed portion. One simple suggestion to alleviate this issue would be to only update $X_g, Y_g$ when there is enough change in local data across enough number of clients, so that such changes are not easily exploitable.

- As a way to strengthen privacy protection under MAFL (and possibly to help with issues mentioned above), we in the server can average within entries of $X_g, Y_g$. If this additional average is done across every random $m$ entries at the server, it would effectively provide averaged data across all local data of $m$ clients, but would result in an $m$-fold decrease in the number of averaged data. This variant is considered in Appendix J.

- In case where the global server is not credential, the averaged data itself should ensure privacy as it is sent to the server. A most obvious concern comes from when $M_k$ is not large enough, so that each entry of $M_k$ reveals more of information of each individual input. Simply using a sufficiently large value of $M_k$ can alleviate this issue, although this might result in worse performance.

- However, for clients whose $n_k$ is quite small, there is a limit for $M_k$ to be large enough. One way to alleviate this issue is to introduce a cut-off threshold for allowing clients to send averaged data to server. We report the results in Appendix H.

**Communication cost**    Since MAFL requires sending averaged input data between server and clients, additional communication costs are incurred. However, it turns out that this additional cost is very small compared to communication cost required for exchanging model parameters. This is mainly due to the fact that input dimension is typically much smaller than number of model parameters. Specifically, for input dimension $d_i$, exchange of averaged data among $N$ clients incurs $2Nd_i$ cost (factor of 2 for server receiving and sending the values). Meanwhile, the cost for exchange of model parameters is $2Np_m$ where $p_m$ is number of model parameters. Under typical circumstances, averaged data is only exchanged at the beginning of the first communication round, while model parameters have to be exchanged every round. Thus the ratio between the two costs after $T$ communication rounds is $d_i/(Tp_m)$. Since $d_i \ll p_m$ in general, we consider extra communication burden to be negligible (even in the worst case where we update averaged data every round, the ratio is still $d_i/(p_m)$.

FedMix also requires calculation of input derivative term in its loss function, so potentially extra memory is required. We further provide additional computation costs of MAFL in Appendix G.

## 4 EXPERIMENTS

We test our result on various benchmark datasets with NaiveMix (direct mixup between local data and averaged data) and FedMix, then compare the results with FedAvg (McMahan et al., 2017) and FedProx (Li et al., 2020b), as well as other baseline Mixup scenarios. We create a highly non-iid environment to show our methods excel in such situations.

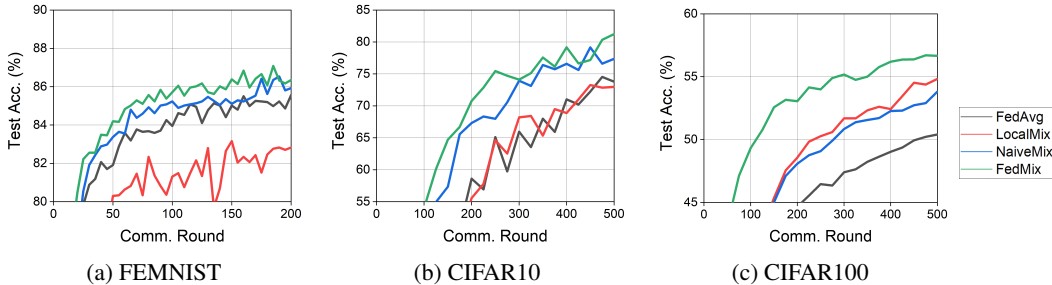

(a) FEMNIST        (b) CIFAR10        (c) CIFAR100

Figure 2: Learning curves for various algorithms on benchmark datasets. Learning curves correspond to results in Table 1. (For simplicity, we only show key algorithms to compare.)

Table 1: **Test accuracy after (target rounds)** and **number of rounds to reach (target test accuracy)** on various datasets. Algorithms in conjunction with FedProx are compared separately (bottom). MAFL-based algorithms are marked in bold.

| Algorithm | FEMNIST | | CIFAR10 | | CIFAR100 | |
|---|---|---|---|---|---|---|
| | test acc. (200) | rounds (80%) | test acc. (500) | rounds (70%) | test acc. (500) | rounds (40%) |
| Global Mixup | 88.2 | 8 | 88.2 | 85 | 61.4 | 54 |
| FedAvg | 85.3 | 26 | 73.8 | 283 | 50.4 | 101 |
| LocalMix | 82.8 | 28 | 73.0 | 267 | 54.8 | 91 |
| **NaiveMix** | 85.9 | 23 | 77.4 | 198 | 53.8 | 85 |
| **FedMix** | **86.5** | **18** | **81.2** | **162** | **56.7** | **34** |
| FedProx | 84.6 | **29** | 77.3 | 266 | 51.2 | 79 |
| FedProx + LocalMix | 84.1 | 39 | 74.1 | 314 | 54.0 | 90 |
| FedProx + **NaiveMix** | 85.7 | 37 | 76.7 | 230 | 53.1 | 74 |
| FedProx + **FedMix** | **86.0** | 32 | **78.9** | **223** | **54.5** | **63** |

## 4.1 Experimental Setup

**Dataset** We implement the typical federated setting where clients have their own local data and one centralized server that receives and sends information from/to the clients. We utilize a large number of clients and only utilize partial set of clients chosen each round to locally update. We used three popular image classification benchmark datasets: FEMNIST (Caldas et al., 2019), CIFAR10, and CIFAR100, as well as a popular natural language processing benchmark dataset, Shakespeare. See Appendix B for more details about dataset, models, and hyperparameters used. We introduce data size heterogeneity for FEMNIST dataset: each client has different size of local data, each from a unique writer. Meanwhile, we introduce label distribution heterogeneity for CIFAR datasets, with clients having data with only a limited number of classes.

**Algorithms** We study the performance of FedMix and NaiveMix and compare with FedAvg and FedProx. We also compare our method against FedAvg with Mixup within local data (labeled LocalMix; see Figure 1(b)), to show whether Mixup within local data is sufficient to allow the model to perform well on external data. To show the effectiveness of FedMix, we also compare our method to the case where we perform direct Mixup with external data (and thus violating privacy, labeled Global Mixup; see Figure 1(a)).

## 4.2 Performance of FedMix and NaiveMix on Non-IID Federated Settings

We compare the learning curves of each method under the same federated settings, in terms of number of communication rounds conducted. Comparing MAFL-based algorithms, NaiveMix shows slight performance increases than FedAvg, FedProx, and Localmix, while FedMix outperforms and shows faster convergence than all of FedAvg, FedProx, Localmix and NaiveMix for all datasets tested as in Figure 2.

While NaiveMix and FedMix is already superior to FedProx, they are parallel to FedProx modification and can be applied in conjunction with FedProx. We compare performances across FedProx

Table 2: Test accuracy after 50 rounds on Shakespeare dataset.

| Algorithm | Global Mixup | FedAvg | FedProx | LocalMix | NaiveMix | FedMix |
|---|---|---|---|---|---|---|
| Test Acc. (%) | 54.4 | 54.7 | 54.4 | 53.7 | **56.9** | **56.9** |

Table 3: Test accuracy on CIFAR10, under varying number of clients ($N$). Number of samples per client is kept constant.

| # of Clients($N$) | 20 | 40 | 60 |
|---|---|---|---|
| Global Mixup | 86.3 | 89.2 | 88.2 |
| FedAvg | 65.8 | 73.4 | 73.8 |
| LocalMix | 46.9 | 71.4 | 73.0 |
| NaiveMix | 62.2 | 75.1 | 77.4 |
| FedMix | **68.5** | **76.4** | **81.2** |

Table 4: Test accuracy on CIFAR10, under varying number of local data per client. Number of clients ($N$) is kept constant. Number of data is indicated in percentage of the case where all 50,000 data are used.

| Local data (%) | 20 | 50 | 100 |
|---|---|---|---|
| Global Mixup | 71.4 | 86.1 | 88.2 |
| FedAvg | 61.8 | 74.7 | 73.8 |
| LocalMix | 43.7 | 60.3 | 73.0 |
| NaiveMix | 51.5 | 69.6 | 77.4 |
| FedMix | **65.2** | **77.8** | **81.2** |

variants of various Mixup algorithms in Table 1. FedMix outperforms vanilla FedProx for various datasets, although they do fall short of default version of FedMix used for the main experiment.

To confirm whether received information is properly incorporated, we compare FedMix with possible Mixup scenarios under MAFL. We show the results in Appendix D.

While Mixup is usually performed for image classification tasks, it could be applied for language models. For language datasets, since Mixup cannot be performed on input, we perform Mixup on embeddings (for a detailed explanation of Mixup between hidden states, see Appendix E). When tested on Shakespeare dataset, FedMix and NaiveMix both show better performance than baseline algorithms (Table 2). Note that for this task, LocalMix has the lowest performance, and global Mixup does not result in the superior performance above federated algorithms as expected. We think Mixup does not provide performance boost for this specific task, but claim that MAFL algorithms still result in better performance compared to FedAvg.

We also claim that FedMix is superior compared to other methods under various settings, in terms of varying number of clients ($N$) and varying number of local data per clients. We observe superior performance of FedMix compared to other algorithms for all settings (see Tables 3 and 4). We also vary the number of local epochs ($E$) between global updates, and still observe that FedMix outperforms other methods (see Appendix F).

**FedMix compared to global Mixup with fixed mixup ratio**   Since FedMix approximates loss function of global Mixup for fixed value of $\lambda \ll 1$, we can evaluate the efficiency of approximation by comparing between FedMix and a global Mixup scenario with fixed $\lambda$ value. Table 5 shows varying performance between global Mixup and FedMix under various values of $\lambda$. As $\lambda$ increases, Mixup data reflects more of the features of external data, resulting in better performance in case of global Mixup. However, this also results in our approximation being much less accurate, and we indeed observe performance of FedMix decreasing instead. The result shows that the hyperparameter $\lambda$ should be chosen to balance between better Mixup and better approximation. However, it seems that high $\lambda$ results in significant decrease in both methods, probably due to external data (which is out-of-distribution for local distribution) being overrepresented during local update.

Table 5: Test accuracy on CIFAR10, under varying mixup ratio $\lambda$.

| $\lambda$ | 0.05 | 0.1 | 0.2 | 0.5 |
|---|---|---|---|---|
| Global Mixup | 79.4 | 80.4 | 81.1 | 63.6 |
| FedMix | 81.2 | 80.5 | 77.7 | 67.1 |

| | $M_k$ | 5 | 10 | 20 | 50 | All |
|---|---|---|---|---|---|---|
| **FEMNIST** | NaiveMix | 85.7 | **86.3** | 86.2 | 86.1 | 85.9 |
| | FedMix | **86.0** | 85.7 | **86.4** | **86.2** | **86.5** |
| **CIFAR10** | NaiveMix | 79.6 | 77.9 | 79.1 | 77.1 | 77.4 |
| | FedMix | **81.4** | **79.9** | **80.4** | **79.5** | **81.2** |

Figure 3: Performance of MAFL-based algorithms for various $M_k$ values (left), and samples of averaged images from EMNIST/CIFAR10 for various $M_k$ values (right).

Table 6: Test accuracy after 500 rounds on CIFAR10, under varying number of classes per client.

| | | class/client | | |
|---|---|---|---|---|
| **Algorithm** | 2 | 3 | 5 | 10 (iid) |
| Global Mixup | 88.2 | 90.7 | 90.9 | 91.4 |
| FedAvg | 73.8 | 84.2 | 86.8 | 89.3 |
| Localmix | 73 | 83.3 | 86.4 | 89.1 |
| NaiveMix | 77.4 | 84.5 | 87.7 | **89.4** |
| FedMix | **81.2** | **85.1** | **87.9** | 89.1 |

Table 7: Test accuracy after 500 rounds on CIFAR10, under varying number of clients trained per communication round.

| | | | $K/N$ | | |
|---|---|---|---|---|---|
| **Algorithm** | 0.1 | 0.15 | 0.25 | 0.5 | 1.0 |
| Global Mixup | 89.3 | 89.7 | 88.2 | 91.2 | 90.7 |
| FedAvg | 63.3 | 73.2 | 73.8 | 76.3 | 83.1 |
| Localmix | 64.7 | 64.5 | 73 | 77.9 | 79.8 |
| NaiveMix | 73.6 | 74.7 | 77.4 | 81.4 | 83.5 |
| FedMix | **74.7** | **76.9** | **80.5** | **82.1** | **84.3** |

**Effect of $M_k$ to compute mean** In our algorithm, we chose to calculate $\boldsymbol{X}_g, \boldsymbol{Y}_g$ with all local data for each client. To observe a potential effect of $M_k$, we varied $M_k$ used to compute the averaged data that is sent from other clients. Inevitably, reducing $M_k$ will result in $\boldsymbol{X}_g, \boldsymbol{Y}_g$ having much more rows, imposing additional computation burden and less preservation of privacy. In general, for both FEMNIST and CIFAR10, there is only small performance decline as privacy is enhanced, as can be seen in Figure 3. We show that using all local data to calculate each mean is sufficient to both preserve privacy and still have good performance.

**Mixup between hidden states** Manifold Mixup (Verma et al., 2018) was proposed to show improvements over input Mixup (Zhang et al., 2018) in various image classification tasks such as CIFAR10, CIFAR100, and SVHN. We discuss the possibilities and implications of applying Mixup between hidden states in Appendix E. In summary, we show that variants of using hidden states do not show meaningful advances over FedMix using input Mixup, suggesting that in general, it is relatively inefficient since it imposes additional communication burden.

**Effect of non-iid-ness and client participation** We claim that our method is efficient when faced with non-iid federated settings. For example, our setting of CIFAR10 having only data from 2 classes per client is very non-iid, as in average a pair of clients share only roughly 20% of data distribution. We test settings for CIFAR10 where clients have data from greater number of classes, and while there is little difference for iid (10 class/client) setting, we observe that FedMix outperform other methods and suffer less from increased heterogeneity from highly non-iid settings (Table 6). In addition, we also observe less decline and better performance for MAFL-based algorithms, FedMix in particular, as we train less number of clients per round, reducing communication burden in cost of performance (Table 7).

## 5 CONCLUSION

We proposed MAFL, a novel framework, that exchanges *averaged* local data, to gain relevant information while still ensuring privacy. Under the new framework, we first suggested NaiveMix, which is a naive implementation of Mixup between local and received data. More interestingly, we proposed FedMix, which provides approximation of global Mixup only using averaged data. MAFL, and FedMix in particular, showed improved performance over existing algorithms in various benchmarks, particularly in non-iid environments where each client has data distributed heterogeneously. While our method is very effective and still preserving privacy, future work needs to be done to deal with various non-iid environments, desirably with better privacy and beyond image classification tasks.

ACKNOWLEDGMENTS

This work was supported by the National Research Foundation of Korea (NRF) grants (No.2018R1A5A1059921, No.2019R1C1C1009192) and Institute of Information & Communications Technology Planning & Evaluation (IITP) grants (No.2017-0-01779, XAI, No.2019-0-01371, Development of brain-inspired AI with human-like intelligence, and No.2019-0-00075, Artificial Intelligence Graduate School Program(KAIST)) funded by the Korea government (MSIT).

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

# A ALGORITHMS

We present a brief depiction of FedAvg in Algorithm 3.

---

**Algorithm 3:** FedAvg

---

**Input:** $N, T, K, E, B, p_k, \mathbb{D}_k = \{\boldsymbol{X}_k, \boldsymbol{Y}_k\}, k = 1, \ldots, N, \eta_t, t = 0, \ldots, T-1$     $LocalUpdate(k, \boldsymbol{w}_t)$:

Initialize $w_0$ for global server            $\boldsymbol{w} \leftarrow \boldsymbol{w}_t$

**for** $t = 0, \ldots, T-1$ **do**                **for** $e = 0, \ldots, E-1$ **do**

    $\mathbb{S}_t \leftarrow K$ clients selected at random        Split $\mathbb{D}_k$ into batches of size $B$

    Send $\boldsymbol{w}_t$ to clients $k \in \mathbb{S}_t$             **for** $batch(\boldsymbol{X}, \boldsymbol{Y})$ **do**

    **for** $k \in \mathbb{S}_t$ **do**                  $\boldsymbol{w} \leftarrow \boldsymbol{w} - \eta_{t+1} \nabla\ell(f(\boldsymbol{X}; \boldsymbol{w}), \boldsymbol{Y})$

      $\boldsymbol{w}_{t+1}^k \leftarrow LocalUpdate(k, \boldsymbol{w}_t)$        **end**

    **end**                             **end**

    $\boldsymbol{w}_{t+1} \leftarrow \frac{1}{K} \sum_{k \in \mathbb{S}_t} p_k \boldsymbol{w}_{t+1}^k$         **return** $\boldsymbol{w}$

**end**

---

# B EXPERIMENTAL DETAILS

**FEMNIST**    FEMNIST is EMNIST (Cohen et al., 2017), a handwritten MNIST dataset, organized into federated setting, as in Caldas et al. (2019). EMNIST is very similar to MNIST, but has several differences. It includes all 26 capital and small letters of alphabet as classes along with numbers, making it 62 classes in total to classify. Also, each image contains information of the writer of the letter. In a realistic non-iid setting, each client has local data consists of only one writer, which is about 200 to 300 samples per client in average, with differing number of samples. We use $N = 100$ clients and trained only $K = 10$ clients per communication round.

We used LeNet-5 (Lecun et al., 1998) architecture for client training. LeNet is consisted of 2 conv layers followed by 2x2 maxpool layer then 3 fc layers. We used 5x5 conv layers with 6 and 16 channels. Following fc layers have exactly the same hidden dimension of original LeNet-5 model.

**CIFAR10 and CIFAR100**    CIFAR10 and CIFAR100 are very popular and simple image classification datasets for federated setting. Both contain 50,000 training data and 10,000 test data. We split the data into each client, $N = 60$ in case of CIFAR10 and $N = 100$ in case of CIFAR100. To create an artificial non-iid environment, we allocate data such that each client only has data from 2 (20 for CIFAR100) randomly chosen classes. We train only $K = 15$ clients per round for CIFAR10 and $K = 10$ for CIFAR100. No validation data was split and we used all training data for local training.

We used modified version of VGG architecture (Simonyan & Zisserman, 2015). Modified VGGnet is consisted of 6 convolutional layers with 3 max pooling layers. 3x3 conv layers are stacked and 2x2 maxpool layer is stacked after every 2 conv layers. Conv layers have channel sizes of 32, 64, 128, 128, 256, 256. Then 3 fc layers are stacked with hidden dimension 512. We use Dropout layer three times with probability 0.1 after the second, third maxpool layers and before the last fc layer. We remove all batch normalization layers since it is reported that they hurt federated learning performance (Hsieh et al., 2020).

**Shakespeare**    We use dataset from *The Complete Works of William Shakespeare*, which is a popular dataset for next-character prediction task. We partition the dataset so that each client has conversations of one speaking role, as in Caldas et al. (2019), which naturally results in a heterogeneous setting, as in FEMNIST. We use $N = K = 60$, each with different number of data (minimum is 200). Since input-level Mixup cannot be performed for discrete character labels, we performed Mixup on the embedding layer. Additional concerns for this variation is considered at Appendix E.

We used 2-layer LSTM, both with hidden dimension of 256. The recurret network is followed by an embedding layer. The output of LSTM is passed to a fully-connected layer, with softmax output of one node per character. There are 84 characters used in the dataset.

Local clients are trained by SGD optimizer with learning rate 0.01 and learning decay rate per round 0.999. We set local batch size as 10 for training. Specific hyperparameter setting for each dataset is

Table 8: Hyperparameter settings for each dataset.

| dataset | FEMNIST | CIFAR10 | CIFAR100 | Shakespeare |
|---|---|---|---|---|
| local epochs ($E$) | 10 | 2 | 10 | 2 |
| local batch size | 10 | 10 | 10 | 10 |
| class per clients | - | 2 | 20 | - |
| fraction of Clients ($K/N$) | 0.1 | 0.25 | 0.1 | 1 |
| total dataset classes | 62 | 10 | 100 | 84 |
| $\lambda$ (for NaiveMix) | 0.2 | 0.1 | 0.1 | 0.1 |
| $\lambda$ (for FedMix) | 0.2 | 0.05 | 0.1 | 0.1 |
| $\mu$ (for FedProx) | 0.1 | 0.1 | 0.01 | 0.001 |

explained in following Table 8. Throughout the experiment, $M_k$ is fixed to local client's dataset size. Changes in these parameters are indicated, if made, are stated for all experiments. Note that we use a fixed small value of $\lambda$ for MAFL-based algorithms to show superior performance.

## C    PROOF OF PROPOSITION 1

We demonstrate mathematical proof of Proposition 1 for FedMix.

Starting from Eq. (3), since the loss function is linear in $y$ for cross-entropy loss $\ell$[2], we have

$$\ell\big(f(\tilde{\boldsymbol{x}}),\tilde{y}\big) = (1-\lambda)\ell\Big(f\big((1-\lambda)\boldsymbol{x}_i + \lambda\boldsymbol{x}_j\big),y_i\Big) + \lambda\ell\Big(f\big((1-\lambda)\boldsymbol{x}_i + \lambda\boldsymbol{x}_j\big),y_j\Big). \quad (6)$$

Unlike the original paper, if we assume $\lambda \ll 1$, we can treat this loss as objective loss function for vicinal risk minimization (VRM). Under this assumption, each term in Eq. (6) can be approximated by independent Taylor expansion on the first and second argument of $\ell$, so that we have

$$(1-\lambda)\ell\Big(f\big((1-\lambda)\boldsymbol{x}_i\big),y_i\Big) + (1-\lambda) \times \frac{\partial\ell}{\partial\boldsymbol{x}}\bigg|_{(1-\lambda)\boldsymbol{x}_i,y_i} \cdot (\lambda\boldsymbol{x}_j)$$

$$+\lambda\ell\Big(f\big((1-\lambda)\boldsymbol{x}_i\big),y_j\Big) + \lambda \times \frac{\partial\ell}{\partial\boldsymbol{x}}\bigg|_{(1-\lambda)\boldsymbol{x}_i,y_j} \cdot (\lambda\boldsymbol{x}_j). \quad (7)$$

Since $\lambda \ll 1$, we can ignore the last term in the second row of Eq. (7), which is $\mathcal{O}(\lambda^2)$. We simplify this equation and switch the second term in the first row and the first term in the second row, to finally obtain

$$\ell\big(f(\tilde{\boldsymbol{x}}),\tilde{y}\big) \approx (1-\lambda)\ell\Big(f\big((1-\lambda)\boldsymbol{x}_i\big),y_i\Big) + \lambda\ell\Big(f\big((1-\lambda)\boldsymbol{x}_i\big),y_j\Big) + \lambda\frac{\partial\ell}{\partial\boldsymbol{x}} \cdot \boldsymbol{x}_j. \quad (8)$$

The derivative $\frac{\partial\ell}{\partial\boldsymbol{x}}$ is calculated at $\boldsymbol{x} = (1-\lambda)\boldsymbol{x}_i$ and $y = y_i$. The coefficient of the last term is changed to $\lambda$ from $\lambda(1-\lambda)$ since we are ignoring $\mathcal{O}(\lambda^2)$ terms.

## D    COMPARISON OF FEDMIX WITH BASELINE MIXUP SCENARIOS

Since our MAFL-based algorithms could be considered as VRM, it could be considered as a data augmentation (Chapelle et al., 2000). Thus, it is important to confirm that increased performance from MAFL not only comes from data augmentation but also from relevant information received from other clients. To check whether this is true, we compare FedMix with algorithms where we use either *randomly generated noises* as averaged data for Mixup (labeled Mixup w/ random noise) and where we use only *locally averaged* data for Mixup (labeled Mixup w/ local means).

In Table 9, we observe that if averaged data for MAFL is substituted for randomly generated noise or locally generated images, it does not show the level of performance FedMix is able to show. Thus, we claim that FedMix properly incorporates relevant information received.

---

[2]We implicitly assume the classification tasks. For regression tasks, we can consider the squared loss function and use the equivalent loss that is linear in $y$

Figure 4: Results for variants of Mixup algorithms for Mixup between hidden states. (a) Learning curves for various algorithms with hidden representation Mixup after $k = 2$ layers. (b) Learning curves for FedMix when Mixup is applied after different numbers of layers.

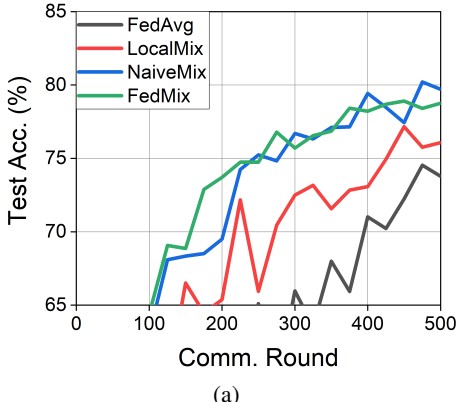

(a)

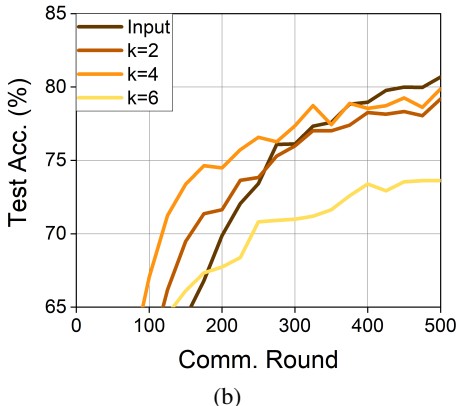

(b)

# E    VARIANT OF FEDMIX AND NAIVEMIX WITH MIXUP BETWEEN HIDDEN STATES

While input Mixup methods promise significant enhancements, one can expect similar performance from hidden state Mixup, originally proposed by Verma et al. (2018). The authors of this work suggest that Manifold Mixup demonstrates similar, if not greater, advantage in terms of performance and adversarial robustness. We can think of variants of FedMix and NaiveMix that implement hidden state Mixup, and test if this variant outperforms vanilla methods based on input Mixup.

Although the original paper proposed randomizing the layer $k$ just before hidden states that undergo Mixup for each batch, we propose setting this layer $k$ constant. This is to reduce communication cost significantly, since selecting randomized layer for Mixup will require other clients having to send multiple hidden states (which usually have large dimensions), further imposing communication burden.

Another change is that while original Manifold Mixup (Verma et al., 2018) suggests backpropagating the entire computational graph through the whole network, including the encoder (part of model projecting input to designated hidden representation) and the decoder (rest of the model projecting hidden representation to output). Such thing is impossible to do in typical federated setting, since the computational graph to calculate hidden representation of local data and the graph to calculate hidden representation of other clients' data is separated, and they cannot be updated simultaneously through local update (doing so requires communicating encoder weights across clients *every* local update, which is highly inefficient in terms of communication). Thus, during Mixup between hidden representations, only the decoder weights can be updated, since only updating encoder of the selected local client will desynchronize encoder weight values for calculating hidden states of local data from those for calculating hidden states of other clients' data every local update, so that the model does not learn properly.

Table 9: Test accuracy after (target rounds) and number of rounds to reach (target test accuracy) on various datasets. We compare FedMix with baseline Mixup algorithms.

| Algorithm | FEMNIST | | CIFAR10 | | CIFAR100 | |
|---|---|---|---|---|---|---|
| | test acc. (200) | rounds (80%) | test acc. (500) | rounds (70%) | test acc. (500) | rounds (40%) |
| **NaiveMix** | 85.9 | 23 | 77.4 | 198 | 53.8 | 85 |
| Mixup w/ random noise | 86.1 | 23 | 77.9 | 201 | 51.2 | 105 |
| Mixup w/ local means | 85.5 | 21 | 73.5 | 233 | 51.0 | 87 |
| **FedMix** | **86.5** | **18** | **81.2** | **162** | **56.7** | **34** |

Table 10: Test accuracy after 500 rounds on CI-FAR10, under varying local epochs ($E$).

| # of Local Epochs ($E$) | 1 | 2 | 5 | 10 |
|---|---|---|---|---|
| FedAvg | 74.4 | 73.8 | 80.7 | 78.9 |
| LocalMix | 63.7 | 73.0 | 74.7 | 80.0 |
| NaiveMix | 72.0 | 77.4 | 81.0 | **82.6** |
| FedMix | **75.8** | **81.2** | **83.0** | 82.5 |

To compensate for this downside, we propose performing vanilla SGD updates without Mixup after Manifold Mixup SGD (which updates weights only in decoder). The vanilla updates will have both local encoder and decoder weights to be updated, thus driving the model to have better hidden representations for Mixup. However, this difference not only imposes additional computation cost, but also does not guarantee that it will show better performance compared to input Mixup methods.

While utilizing hidden representations from other clients sound like a safe idea, it does not ensure data privacy, primarily because the updating client has knowledge of the exact encoder weight values used to calculate hidden states received, and based on our modification, the received hidden states are treated as constants during Mixup. Model inversion attacks (Fredrikson et al., 2015) have been suggested to recover input images from hidden states or outputs, with access to weight values. Thus, direct Mixup between hidden states does not guarantee data privacy. Variant of FedMix and NaiveMix can be applied during decoder training phase, so that privacy is ensured while we successfully approximate Mixup.

The performance of proposed algorithms is shown in Figure 4 on CIFAR10 (same settings with main experiment for dataset, model, and training is used; see Appendix B). Comparison between methods in Figure 4(a) shows that while variants of FedMix and NaiveMix show improved performance over existing methods, they still do not outperform our method based on input Mixup (compare with dotted line). Meanwhile, comparison between using different layers for Mixup is shown in Figure 4(b). It is shown that $k = 4$ has fastest learning curve but converges similarly to case of $k = 2$, both being slightly outperformed by case of input Mixup.

Considering additional computation burden required to communicate hidden states (which often have larger dimensions than raw input) and necessity to communicate the hidden states every communication round (since hidden representations change with encoder weights), we propose that FedMix using input Mixup is superior, and use this method for our main analyses.

## F    EFFECT OF LOCAL EPOCHS

Previous works (McMahan et al., 2017; Caldas et al., 2019) show that number of local epochs, $E$, affects federated learning performance. We tested the effect of $E$ on CIFAR10. In general, we showed that test performance increases as $E$ increases. In addition, we observed that under various values of $E$, FedMix shows the best performance compared to other algorithms (see Table 10), being a close second after NaiveMix for $E = 10$. MAFL-based algorithms outperform existing algorithms for all values of $E$ tested.

## G    ADDITIONAL COMPUTATION COST INCURRED BY MAFL

For FedMix, additional computation and memory are required on edge devices during model training for each communication round, since $\ell_{\texttt{FedMix}}$ requires additional terms, including gradient by input, $\frac{\partial \ell}{\partial \boldsymbol{x}}$, compared to vanilla FedAvg. We claim that FedMix does not result in an additional computation burden. Specifically, we trained FedMix on CIFAR10 with the same settings as the main experiment to 70% accuracy in 1.94 hours; FedAvg takes 1.95 hours. FedMix spends a comparable amount of time to reach a similar level of performance of FedAvg. While in the memory aspect, FedMix requires about twice more GPU memory allocation compared to FedAvg, this phenomenon is also observed on LocalMix and NaiveMix. The extra memory burden comes from Mixup by enlarging the input dimension twice. For instance, FedAvg requires 46.00MB to allocate, LocalMix requires 94.00MB and 98.00MB for FedMix. Calculating gradient of the input derivative gives only negligi-

ble 2-3MB additional memory usage, which is reasonable concerning the substantial performance increase from LocalMix to FedMix.

## H INTRODUCTION OF CUT-OFF THRESHOLD IN MAFL

To better ensure privacy, a cut-off threshold that prevents clients with fewer data to send averaged data could be introduced. We performed this in FEMNIST, since for such procedure to be effective, heterogeneous size of local client data is necessary. We test with $N = 300$ clients, and introduce different threshold levels to test its efficiency. In addition, we also test with multiple $\lambda$ values, to see whether threshold level affects optimal value of $\lambda$ for FedMix.

We present the results in Table 11. While threshold does not hugely affect performance, we observe that a moderately small threshold level of 100 results in the best performance. We suggest that as the threshold level is heightened, there is less overfitting to clients with small size local data, but it also results in a decrease in the number of averaged data received by each client. We indeed find an appropriate value of threshold that maximizes performance.

In case where there are a different number of data per client, the sensitivity of $\lambda$ could also be different compared to when all clients have the same number of data. Results in Table 13 show that there is little change in performance by change in $\lambda$, especially compared to Table 5. In addition, an inspection of the performance of a global model on individual test data of clients does not reveal any noticeable pattern by the size of local data (see Table 13).

Table 11: Test accuracy on FedMix with introduction of cut-off threshold, tested on a number of threshold level. Optimal value of $\lambda$ is also shown.

| $\lambda$ | threshold | | | |
| | 1 | 100 | 150 | 200 |
| --- | --- | --- | --- | --- |
| Test Acc. (%) | 83.0 | **83.3** | 83.1 | 82.9 |
| $\lambda_{\texttt{optimal}}$ | 0.2 | 0.05 | 0.1 | 0.2 |

Table 12: Test accuracy on FEMNIST, $N = 300$ under various Mixup ratio $\lambda$.

| $\lambda$ | 0.05 | 0.1 | 0.2 |
| --- | --- | --- | --- |
| Test Acc(%) | 82.8 | 82.8 | **83.0** |

Table 13: Mean and standard variation of local test accuracy of FedMix on FEM-NIST, tested on clients with varying local data size, under varying $\lambda$.

| $\lambda$ | $n_k < 100$ | | $100 \leq n_k \geq 199$ | | $n_k > 199$ | |
| | mean | std | mean | std | mean | std |
| --- | --- | --- | --- | --- | --- | --- |
| 0.05 | 85.8 | 15.1 | 77.6 | 14.8 | 85.4 | 9.3 |
| 0.1 | 81.1 | 17.4 | 76.4 | 14.1 | 86.2 | 9.5 |
| 0.2 | 83.9 | 16.2 | 77.6 | 14.8 | 85.8 | 10.0 |

## I MAFL IN CONJUNCTION WITH GAUSSIAN NOISE

With results in Figure 3, we expressed concern with small values of $M_k$ causing privacy issues with only a small performance boost, if at all. A common practice of introducing additional privacy is adding Gaussian noise. This is a popular method associated with differential privacy (McMahan et al., 2018), but adding noise alone does not guarantee differential privacy, since the noise level should be explicitly linked to differential privacy levels, $\epsilon$ and $\delta$. Addition of artificial pixel-wise noise will enhance privacy but will result in a quality drop of averaged data. While privacy added by noise and privacy from averaging data cannot be directly compared, we can select a noise level which in conjunction with small $M_k$, visually provides data privacy similar to that of maximum $M_k$.

Results show that the introduction of Gaussian noise does result in a decline in performance (Table 14),although the decline is very small. Interestingly as noise gets larger as $\sigma = 0.3$, random noise

Table 14: Performance of FedMix with Gaussian noise. $\sigma$ refers to standard deviation of Gaussian noise.

| $\sigma$ | 0 | 0.05 | 0.075 | 0.15 | 0.3 |
|---|---|---|---|---|---|
| $M_k = 5$ | 81.4 | 80.1 | 81.1 | 78.7 | 81.5 |
| $M_k = 10$ | 79.9 | 80.8 | 79.1 | 79.4 | 81.7 |
| $M_k = 20$ | 80.4 | 80.5 | 79.7 | 80.7 | 81.0 |

Table 15: Test accuracy on CIFAR10 varying $m$, number of entries in $X_g, Y_g$ to be further averaged.

| $m$ | 1 | 4 | 10 |
|---|---|---|---|
| Test acc (%) | 81.2 | 81.6 | 78.4 |

provides an effect as data augmentation and results in a performance increase compared to $\sigma = 0$. This experiment is in line with Appendix D. We conclude that introduction of noise in averaged data could provide us with a reasonable alternative to FedMix with large $M_k$. While our method does not align directly with differential privacy, we leave as future work how FedMix could be smoothly combined with DP-related methods and how its privacy could be quantified in terms of differential privacy.

## J  ADDITIONAL EXPERIMENTS: VARIATIONS OF FEDMIX

**Averaging within $X_g, Y_g$**  Further averaging between entries of $X_g, Y_g$ practically provides an extension of the range of viable $M_k$ such that it exceeds $n_k$, in the sense that each averaged data is from multiple clients' data. Such a process would also result in fewer data included in $X_g, Y_g$, so we tested effect of this procedure on model performance. Table 15 shows that for $m$-fold extra averaging, we even observe increase in performance, but it quickly declines as $m$ gets too large. This method provides an improvement in privacy while even possibly resulting in better performance.

**Effect of same-class split for averaging**  We perform random split of local data for averaging, but an unbalanced split, such as only averaging data with the same class labels, could result in better performance. We compared between random split and same-class split while keeping $M_k = 0.5n_k$ be equal for both methods. Same-class split resulted in a significant decline in performance, and we conclude that there is no advantage of such split over random split that we are using for our main results.

Table 16: Test accuracy on CIFAR10, with class/client = 2, under different split methods. $M_k = 0.5n_k$ for both splits.

| | random split | class split |
|---|---|---|
| FedMix | **81.2** | 78.8 |

Table 17: Test accuracy of NaiveMix on CIFAR10, under varying Mixup ratio $\lambda$.

| $\lambda$ | 0.05 | 0.1 | 0.2 | 0.5 |
|---|---|---|---|---|
| NaiveMix | 79.5 | 79.9 | 80.6 | 29.8 |

**NaiveMix with varying Mixup ratio $\lambda$**  We varied Mixup ratio $\lambda$ for NaiveMix as well. Results in Table 17 shows that NaiveMix also has an intermediate optimal value of $\lambda$. The drop in performance for $\lambda = 0.5$ is much more dramatic than for FedMix (see Table 5 for comparison with Global Mixup and FedMix). We think that NaiveMix loss also suffers as it gives more weight to the averaged data, especially for large $M_k$.

**Heterogeneity from skewed label distribution**  Recent papers (Yurochkin et al., 2019; Wang et al., 2020) suggested an alternative heterogeneous environment, which does not limit the number of classes per client but skews label distribution in local data. We used a Dirichlet distribution of $\alpha = 0.2, 0.5$ as described by Yurochkin et al. (2019) and Wang et al. (2020). Results show that FedMix still outperforms all other algorithms. We think that such label skewing introduces less heterogeneity compared to our practice of limiting the number of classes per client, but nevertheless, FedMix is still the most powerful method in terms of performance.

Table 18: Test accuracy on CIFAR10 under label-skewed heterogeneous environment. We used Dirichlet distribution for uneven label distribution.

| $\alpha$ | FedAvg | GlobalMix | LocalMix | NaiveMix | FedMix |
|---|---|---|---|---|---|
| **0.2** | 83.9 | 91.1 | 84.0 | 85.0 | **86.4** |
| **0.5** | 87.6 | 91.1 | 88.0 | 88.2 | **88.4** |

