# OpenReview forum: "FedMix: Approximation of Mixup under Mean Augmented Federated Learning"
_ICLR.cc/2021/Conference — ICLR 2021 Poster_

### Official Review · AnonReviewer4 · 2020-10-20
**Interesting work, good ideas and evaluation. Some aspects need to be improved.**

**Rating:** 7
**Confidence:** 4

**Review:**

In this work, the authors aim to approach the non-iid data issue in FL by allowing for mean of the local client data to be transmitted in addition to the model parameters. I find this work very interesting and the paper well executed.
First, the authors present the logic for MAFL, which encompasses the sending and receiving of other clients' averaged data, followed by FedMix, a method for augmenting the local data-set with the averaged data from other clients.

Throughout the method section and their experiments, the authors show the benefits of MAFL+FedMix by ablation to other MixUp inspired approaches.

My issues with this paper are along some different aspects:
Privacy:
Sending statistics of local data is inherently less private than sending model parameters alone. The authors mention this explicitly, but do not go into more detail. I understand that the notion of privacy in FL is a research topic in itself, but I would wish for a more nuanced discussion of the trade-offs here. Throughout the experiment section, the largest 'federation' of devices is N=100 for Cifar100 and Femnist. Taking cifar100 as example, each client has 50k/100 = 500 data-points, the average of which I can agree intuitively to be not very informative (at least visually) and the 'discriminative information' that the authors mention, is presumably not very high. However, 500 data-points can still be considered a large amount of data-points for the federated scenario. As the number of data-points per client $n_k$ decreases, the more information about individual data-points is contained in their average. The problem is increased as $M_k>1$ . Further, 'discriminative information' is not the only privacy-worthy information in FL. Differential Privacy, for example, is trying to quantify if an individual data-point is present in a local data-set. Since a client receives a concatenation $(X_g,Y_g) = ({\bar{x}_1,\bar{x}_2,...,\bar{x}_N},{\bar{y}_1,\bar{y}_2,...,\bar{y}_N})$ of all clients' averaged data-sets, an individual client's participation in the training can also not be hidden from other clients.
Furthermore, the formulation in Algorithm 1 implicitly assumes a continual learning setup where clients might be collecting more data as training progresses. In its current formulation, the authors do not mention if the batches are re-computed randomly, opening up the possibility for attacks on the differences between batches across time.

Computational Burden:
FedMix requires computing gradients through the Taylor expansion (EQ 4), which increases computation and memory requirements. Especially in a federated setting, computation and memory are constrained resources, so I would expect the authors to provide some estimates over the additional requirements for computing gradients $\nabla_w l_{FedMix}$

Experimental Evaluation:
I am missing some details on the setup for the FEMNIST dataset. At the moments, the authors mention selecting 100 clients, however I wonder if they used the writer-id or re-shuffled to create a controlled label-skew. If they used the writer-id, how did they select the subset of 100 clients?

Some details:
I believe in Figure 1 b), the indices above 'Local data' should be $i$, not $j$.
Directly below Figure 1, the sentence should begin with: "A more practical approach to..."
Algorithm 2 could be improved, I believe. I see no space constraint that would prevent including some more detailed information analogous to Algorithm 3 in the Appendix. I am assuming the gradient is calculated mini-batch wise. Ideally, the $LocalUpdate$ would receive the same arguments as those that it is being called with on the server side for example.
Top of page 5: 'meshed' -> 'mashed'.
Just above Eq (2): '... client i has access to ...' (remove 'an').


The experiments that would make this evaluation great in my opinion:
Train on the full FEMNIST set of 3600clients including all those clients with very small number of data-points. Then introduce a cut-off-threshold for the minimum number of data-points that each client has to have in order to send its averaged data to the server. Alternatively, add random noise to these averages in relation to how much data is present. There is probably a differential privacy formulation that would make the required noise-level explicit. This noise level or cut-off-point should give more insight on several dimensions of the proposed work:

- How sensitive is FedMix to different minimum required data-points as a trade-off with privacy.
- How sensitive is $\lambda$ to different number of data-points per client (or the consequence of fixed $\lambda$ generally as number of data-points per client differs). Since no other experiment has different number of data-points per client, I believe this to be relevant.

Additionally, to further increase privacy, the authors might consider (randomly) averaging some of the elements in $(X_g,Y_g)$ before sending the data to clients and study those effects.

Summarizing, I want to thank the authors for this very interesting read and interesting insights. If the authors provide a more nuanced/detailed discussion of the privacy aspects of their work and extend their experimental section with the more holistic FEMNIST experiment I described above, I will raise my score! I see no violation of the CoE in this work.

Finally, I cannot believe that the authors let the opportunity slide to name their algorithm 'FedUp' ;)

---

> ### Author Response · Authors · 2020-11-20
> **Response to AnonReviewer4's Review**
>
> 7.Train on the full FEMNIST set of 3600clients including all those clients with very small number of data-points. Then introduce a cut-off-threshold for the minimum number of data-points that each client has to have in order to send its averaged data to the server. ...
>
> We thank the reviewer for suggesting a nice holistic experiment. We are currently running the experiment with $N=3600$ clients, but sadly, we think this will take too much time, from about several days to several weeks. Thus, we first briefly present test results with $N=300$ clients (we had $N=100$ for our main FEMNIST result). We think $N=300$ will not result in a significant difference, since we do have a sufficient number of clients that have few local data, but if you have concerns and think setting $N=3600$ is necessary, please remind us.
>
> We introduced various cut-off threshold values to see the effect of constraining clients' updates to the server. In the most extreme case, a threshold of 200 excludes about half of all clients from sending their averaged data. For each threshold, we also test various $\lambda$ values to assess the model's sensitivity to $\lambda$. We added the details of this new experiment as well as the results in the revision.
>
> To summarize, the threshold of 100 (excluding <10% of clients) resulted in the best possible performance (even better than the case without the cut point). We think that excluding clients with substantially fewer data than others could improve performance since there is less overfitting to such local clients' data. However, a further increase in the threshold resulted in a decline in performance, likely because there is fewer data to Mixup with, reducing its effectiveness.
>
> The reviewer also suggests to observe sensitivity of $\lambda$ to different number of data per client.  We evaluated performance for various values of $\lambda$ without using cut off option here. We observe that $\lambda=0.2$ is the optimal value (same as for our $N=100$ main experiment), although there is only a small drop in performance for $\lambda=0.05,0.1$ as each results in only 0.2% less test accuracy. Comparing against CIFAR-10 in Table 5 (where all clients have same number of data), the performance of the model in this experiment is less sensitive to value of $\lambda$. Meanwhile, we also evaluated for each client that has different number of data in this experiment. We observe that there is no particular tendency on the sensitive to $\lambda$ according to the number of data. This experiment was also added on Appendix in the revision.
>
> 8.Additionally, to further increase privacy, the authors might consider (randomly) averaging some of the elements in $(X_g,y_g)$ before sending the data to clients and study those effects.
>
> Thank you for suggesting a very interesting extension. Averaging among $(X_g,y_g)$ as suggested by the reviewer practically can provide additional protection of privacy. This procedure will further increase privacy but will result in less number of data points in the set $(X_g,y_g)$ for training.
>
> We added the results of this additional experiment in the appendix of the revision. First we define a parameter $m$ indicating how much entries of $(X_g,y_g)$ were used for further random averaging. On CIFAR-10, for $m=4$, FedMix results in an accuracy of 81.6%, which is even higher than our main result of 81.2%, although not by much. Meanwhile, $m=10$ results in a significant decrease to 78.4%. We think that optimal value of $m$ is decided by balancing between higher privacy (larger $m$) and more data points in $(X_g,y_g)$ (smaller $m$). We think that this averaging among $(X_g,y_g)$ opens the possibility of averaged data come from more than $n_k$ individual data, further ensuring privacy, and potentially even increasing performance.

---

> ### Author Response · Authors · 2020-11-20
> **Response to AnonReviewer4's Review**
>
> 4.**Computational Burden** imposed by calculating gradient by input
>
> We agree with the reviewer's concern about additional computational resources. However, it turns out that FedMix does not impose a huge computational burden on local clients. To verify this, we measured the computation time and memory usage on CIFAR-10 under the same setting as our main experiment. Throughout the measurement, we computed the number of computational resources in the same device. In this experiment, a single round of FedMix takes 43.1s which is twice the time spent by FedAvg, 24.75s. Considering the multiple communication rounds until reaching a certain performance, the time difference is reduced. Specifically, for FedMix, it takes 162 communication rounds and 1.94 hours in total to reach 70% test accuracy while FedAvg takes 283 communication rounds and 1.95 hours in total to attain the same test accuracy.
>
> We also measured GPU memory usage of FedAvg, LocalMix, and FedMix. FedAvg uses at most 46.00MB of allocated memory and 27.48MB of active memory. As LocalMix doubles the input dimension to use Mixup, LocalMix results in 94.00MB of allocated memory and 75.40MB of active memory, which is double the memory of FedAvg. However, the memory difference between LocalMix and FedMix is surprisingly negligible considering the substantial performance increase in FedMix. FedMix uses 98.00MB allocated memory and 79.92MB active memory, with only a significantly small 2-3MB difference compared with LocalMix. Despite having to obtain an additional derivative of loss, FedMix only needs a small additional amount of memory compared to LocalMix.
>
> **Experimental Evaluation**:
>
> 5.What is used to create data heterogeneity in FEMNIST? If writer-id, how the subset is chosen?
>
> We had included a detailed description of experimental settings in Appendix B in the original submission. We used writer-id to create the non-iid environment. In fact, each client has one writer-written data. Although most writers have data points of size within 200~400, we do have a few clients that have even less than 50 data points. We selected a subset of 100 writer-ids randomly from the entire list. Thus, our FEMNIST experiment is actually under an environment with a different number of data points per client, with $M_k$ being equal to the size of local dataset of each.
>
> 6.Some Details: ...
>
> Thank you for the suggestions. We fixed all the minor errors that the reviewer pointed out. We also revised Algorithm 2 following the reviewer's advice.

---

> ### Author Response · Authors · 2020-11-20
> **Response to AnonReviewer4's Review**
>
> We thank the reviewer for a thorough and insightful review. We address the concerns below:
>
> **Privacy**: We do agree with the reviewer that privacy aspects in federated learning are very difficult to quantify and are highly dependent on the target applications as well. But, we are happy to discuss more thoroughly the aspect of privacy since MAFL requires exchanging additional data, ensuring privacy is surely a concern. We summarize the discussion points brought out by the reviewer and comment on each. We also have added a section with discussions about issues on privacy including ones raised by the reviewer and possible improvements in the revision.
>
> 1.As the number of data-points per client decreases, more information about individual data points is contained in their average. The problem is increased as $M_k>1$.
>
> We thank the reviewer for pointing out this issue. We elaborated on this issue and discussed some possible solutions in the revision. Specifically, one simple way to address this issue is to introduce a cut-off threshold for clients *not* to send their averaged data to the server, as suggested by the reviewer in the holistic experiment below. This simply prevents such clients from causing privacy concerns, but it could cause a decline in performance. Another way is to simply merge averaged local data from a number of clients at the server before distribution (assuming that the server is completely trusted). There would be an inevitable trade-off between the number of entries in the set $X_g$ instead, so this could also result in a decline in performance.
>
> 2.Since a client receives a concatenation of all clients' averaged datasets, an individual client's participation in the training can also not be hidden from other clients.
>
> The reviewer points out that from averaged data, its client ownership could be identified and thus each client's participation in training is revealed to other clients. We do not think such identification can be easily done. Since the client only receives averaged data, and no other information of a specific client such as client-id, we think it is very difficult to figure out client participation from just averaged data from all clients, particularly in the case of large $M_k$. If this is not the exact concern of the question, please clarify us of the concern.
>
> 3.The authors do not mention if the batches are re-computed randomly, opening up the possibility for attacks on the differences between batches across time.
>
> Thank you for giving us insightful comments. Honestly, we did not take such risk into account, but fortunately, we do perform a random split for every update of $X_g$ and $y_g$ (See Algorithm 1). Hence, we think it is unlikely to occur, especially in case of $M_k<n_k$, where every update would involve a different split of local data to compute the means.
>
> In order to further reduce this potential risk, we can additionally introduce the following safety guards. When the global server could be trusted: (1) if we are concerned about the possibility of identification of each entry of $(X_g,y_g)$, simply shuffling elements in $(X_g, y_g)$ in the global server, in case where enough number of clients update their data so that matching individual entries is very difficult, can prevent even such attempts to attack on changes in averaged data. Also, (2) each client does not have any information about $M_k$ of other clients, so attacks that require knowledge of $M_k$ of a client is defensible. Meanwhile, if privacy of data to the global server is a concern, (3) instead of updating $(X_g, y_g)$ every shift, we can deter updates until there is enough amount of change of local data over a certain threshold value. Under this modification, we can reduce the possibility of identification of individual data by comparing data before and after updates.

---

> ### Comment · AnonReviewer4 · 2020-11-23
> **Thx for the rebuttal**
>
> Thx for addressing my concerns.
> The only remaining feedback I have is wrt. to the 'DP' experiments.
> Differential privacy crucially is not only about empirical results. Instead, DP provides strict privacy guarantees based on proving how an algorithm is $\epsilon,\delta$ differentially private. While I commend the authors of including experiments with noised-up communications, this is by no means 'differentially private'.
> I raise my score to 7 under the condition that the authors remove these claims or provide a proof about the DP nature of their algorithm.
>
> Furthermore, I saw some sentence structure errors in the added 'red' text, so for a camera ready version, might I suggest another proof-read.

---

> > ### Author Response · Authors · 2020-11-23
> > **Comment for Comment**
> >
> > Dear AnonReviewer4,
> >
> > We really thank you for your helpful and detailed feedback.
> >
> > While Gaussian noise is a typical practice in differential privacy literature, we agree that it does not apply exactly to the privacy provided by our algorithm. We revised the related section accordingly in our new revision.
> >
> > We welcome your suggestion to proofread our newly added sections!
> >
> > Please let us know if there is anything else that you want to discuss further.
> >
> > Thank you again for your time and please keep safe!

---

> > > ### Comment · AnonReviewer4 · 2020-11-23
> > > **About DP**
> > >
> > > For completeness sake: Indeed, Gaussian noise is typical to use in DP, but crucially the magnitude (variance) of the noise makes all the difference. You can add $\sigma$ amounts of noise without knowing *how private* that actually is. This is why you need to rigorously link $\sigma$ (and typically additionally other mechanisms, such as clipping) to the $\epsilon,\delta$ levels of DP.

---

> > > > ### Author Response · Authors · 2020-11-24
> > > > **Comment for Comment on DP and Gaussian Noise**
> > > >
> > > > We understand that while Gaussian noise is typical to use in DP, it alone does not guarantee the link between noise level and differential privacy level $\epsilon, \delta$. We are aware that Gaussian noise does not trivially provide differential privacy, and the level of DP associated with the noise level should be estimated. Hence, following your suggestion, we have thoroughly proofread the section so that it does not contain any misunderstandings (such as `DP-FedMix') regarding differential privacy.
> > > >
> > > > In addition, in our new revision, we did extend the experiment in Appendix I, which adds noise to averaged data, for broader range of $\sigma$ and $M_k$ values. The addition of noise interestingly gives us generally worse performance compared to their counterparts without noise, although some of them slightly outperform them. We think there is a trade-off between the effect of data augmentation and the effect of more accurate averaged data.
> > > >
> > > > We hope we have thoroughly addressed your concerns. Please let us know if you have any remaining concerns.

---

### Official Review · AnonReviewer1 · 2020-10-20
**A novel method, but the experimental justification can be improved**

**Rating:** 6
**Confidence:** 4

**Review:**

##########################################################################

Summary:

The paper proposed MAFL, a novel approach to conduct Mixup under the federated learning setting whiling preserving data privacy. The proposed FedMix scheme is inspired by Taylor’s expansion of the global Mixup formulation. The effectiveness of MAFL is justified via empirical studies over a simulated federated learning environment, which indicates that Mixup achieves better test accuracies on various machine learning tasks.

##########################################################################

Reasons for score:

My overall evaluation score on the current manuscript is borderline reject. The research direction on studying the effectiveness of data augmentation under the federated learning setting is promising. The formulation and motivation of the proposed MAFL scheme are sound. The main justification on FedMix is from the experimental study, which can be further improved e.g. the communication cost and privacy of FedMix can be more explicitly studied. If the proposed MAFL scheme can be supported by some theoretical analysis, the current manuscript can be much stronger. I will be happy to increase my overall evaluation score if my major concerns are addressed.

##########################################################################

Pros:

1. The paper studies the effectiveness and practicality of conducting data augmentation under the federated learning scenario, which is quite promising and can potentially gain impact.

2. The proposed FedMix method is motivated via using Taylor expansion to approximate the global Mixup data augmentation objective, which makes sense in general.

3. Extensive experimental results are provided under the image classification and the next-word prediction tasks under the simulated non-iid environment, which indicates that FedMix enjoys high effectiveness on improving the model test accuracy under the data heterogeneity.

##########################################################################

Cons:

1. The main concern on the proposed FedMix method is communication and computation efficiency. From the proposed Algorithm 1, for each FL round, MAFL requires all available clients to upload their locally averaged data batches. It is easy to imagine in a real federated learning environment (with up to $10^{10}$ available clients), it can lead to a significant communication overhead [1]. Thus, it would be useful to explicitly study the communication cost of MAFL, e.g. report Test Accuracy vs the amount of communication for Figure 2 can help to understand the communication efficiency of MAFL better.
2. It’s not clear how MAFL splits the local datasets. Does it a conduct random split? Would it be possible for each local client to put “similar” (e.g. data points within the same class) into the same batch? Such an approach intuitively preserves more data property in $\bar x$ and $\bar y$.
3. Although a value of small $M_k$ in MAFL leads to worse data privacy, it’s easy to imagine the proposed MAFL can be combined with other differential private (DP) method e.g. [2]. It will be useful to consider a DP version of MAFL.
4. The authors are encouraged to add the baseline of “Global Mixup” to Table 2, 3, 4, 6, 7 to understand the gap between the proposed FedMix method and the “ideal” baseline.
5. The FedProx result on FEMNIST is a bit confusing, what accuracy will FedProx reach for running 32 FL rounds?
6. For the image classification tasks, it seems FedMix outperforms other baselines. However, for the language task e.g. Table 2, it only matches the accuracy of NaiveMix. Does it mean FedMix can be improved for augmenting language examples?
7. The approach to simulate data heterogeneity in the current paper can be generalized by the method proposed in [3-4]. It would be useful to consider the Dirichlet distribution based data partition strategy.

[1] https://arxiv.org/pdf/1912.04977.pdf

[2] https://arxiv.org/pdf/1710.06963.pdf

[3] https://arxiv.org/pdf/1905.12022.pdf

[4] https://arxiv.org/pdf/2002.06440.pdf

#########################################################################

Minor Comments:
1. It seems the CIFAR-10 and CIFAR-100 curves in Figure 2 are not reaching to full convergence. Thus, it would be helpful to run the experiments for more FL rounds.
2. Missing references: [1-2]. And some references are sort of outdated e.g. “Federated optimization in heterogeneous networks” (T Li et al) was accepted to MLSys 2020.


[1] https://arxiv.org/abs/1912.04977

[2] https://arxiv.org/abs/1908.07873


###################### Post Rebuttal #############################
Most of my concerns on the current manuscript are addressed. I tend to increase my overall evaluation score to 6.
#############################################################

---

> ### Author Response · Authors · 2020-11-15
> **Response to AnonReviewer1's Review**
>
> 5.The FedProx result on FEMNIST is a bit confusing, what accuracy will FedProx reach for running 32 FL rounds?
>
> FedProx shows 80.8% accuracy after 32 FL rounds. But as we clearly mentioned in the caption, Table 1 shows the test accuracy after (target rounds) and the required number of rounds to reach (target test accuracy). Hence, round 29 for FedProx in Table 1 does not mean that it achieves 84.6% with 29 rounds, but it means that it took 29 rounds to reach test accuracy of 80%. So the result would be interpreted that while FedProx reached 80% accuracy slightly faster, the final accuracy (with 200 rounds for all methods) was much lower than FedProx+FedMix. We revised the table to make sure the readers understand the data more easily.
>
> 6.Can FedMix be improved for augmenting language models?
>
> While Mixup has been successfully implemented in NLP [1], Mixup is not a popular procedure implemented in most prominent works in NLP. Interestingly, even Global Mixup does not result in an obvious increase in performance compared to FedMix. At some Mixup ratio, Global Mixup sometimes results in performance degradation than FedMix or other baselines. We think that for FedMix to be more effective, Global Mixup should first show more significant improvement over FedAvg, which is not that case in this task. We think that this is the main reason why FedMix and NaiveMix show similar performances, but we want to stress that both FedMix and NaiveMix are the members under our MAFL framework and they are still better compared to other baselines in Shakespeare dataset. We also note that since Mixup on raw input is not possible on the language datasets as it is done for images, we implement a latent Mixup version of each method (discussion about this alternative version on image classification tasks is present in the appendix of the original paper).
>
> 7.Use Dirichlet distribution to introduce data heterogeneity.
>
> We thank the reviewer for providing another good way to introduce data heterogeneity. We conducted additional experiments using Dirichlet distribution with $\alpha=0.5$ to introduce the suggested heterogeneity on CIFAR-10 dataset. To summarize, FedMix still has the best performance with 88.4% accuracy, while FedAvg results in 87.6% accuracy. Since this setting is **less** heterogeneous than our introduction of a limited number of classes per client, there is a smaller difference between the algorithms.
>
> We added the results in the revision in Appendix J.
>
> Minor comments
>
> 1.Convergence of FedMix on CIFAR-10 and CIFAR-100
>
> We ran some of the experiments for more FL rounds. For instance, when tested for 1,000 rounds on CIFAR-10, FedAvg converges to 82.7% accuracy, and FedMix to 84.4% accuracy, outperforming FedAvg.
>
> 2.Missing References
>
> References are added and fixed in our revision.
>
> [1] https://arxiv.org/abs/1905.08941.pdf

---

> > ### Comment · AnonReviewer1 · 2020-11-23
> > **Thanks for the rebuttal**
> >
> > I carefully read the authors' response. The authors addressed most of my concerns by adding additional experimental results as suggested, and MAFL demonstrates good effectiveness. I tend to raise my overall evaluation score from "5" to "6".
> >
> > An additional comment over the communication overhead for MAFL is that the authors are also expected to discuss the communication overheads for input images with high resolutions e.g. [1]. For CIFAR-10 scale, it's clear that the input dimension is lower than the model dimension. But for the images with super high resolution, it's not clear.
> >
> > [1] https://arxiv.org/pdf/1902.06068.pdf

---

> > > ### Author Response · Authors · 2020-11-24
> > > **Comment for Comment on High-Resolution**
> > >
> > > We thank you greatly for your reevaluation.
> > >
> > > We understand the concern that the fact that input dimension is much smaller than number of model parameters is not so obvious for high-resolution image datasets. We should first note that for discriminative models, which we mainly consider in our paper, uses datasets such as ImageNet (size $224 \times 224 \times 3 \approx 150\text{k}$). However, models commonly used for ImageNet, such as VGG16 (138M parameters), ResNet-50 (26M parameters), and even GoogLeNet (5M parameters) have a lot more model parameters than input dimension. So, we argue that additional communication cost is still much smaller than cost required for exchanging model parameters even under assumption that we exchange averaged data every iteration.
> > >
> > > We would also like to emphasize that unlike generative models (the paper cited in your comment also discusses image super-resolution (SR) task and it only handles low resolution images as its inputs), the resolution of input images is not so high in most discriminative models. In addition, it might not fit well into real FL situations, where edge devices have limited amount of computational resources available. We could still consider exchanging averaged data and model parameters for image SR tasks. For example, Set5, a widely used image SR dataset, has about $313 \times 336 \times 3 \approx 316\text{k}$ parameters for its high-resolution version, but this high resolution image is used as output for image SR tasks, as mentioned. Some generative models nowadays also deal with high resolution images such as FFHQ (again note that it is used as output), and in this case, the cost is $1024 \times 1024 \times 3 \approx 3.14\text{M}$. This could make additional communication cost for some models considerable, but if we consider that exchanging averaged data typically only happens in the beginning, its costs will be reduced enough as there are more FL rounds.

---

> ### Author Response · Authors · 2020-11-15
> **Response to AnonReviewer1's Review**
>
> We thank the reviewer for a thorough and insightful review. We address the issues pointed out by the reviewer below:
>
> 1.Additional communication burden imposed by MAFL.
>
> We believe that our MAFL framework incurs negligible additional communication cost compared to the cost incurred by exchanging model parameters for FL situations. First of all, please note that the additional exchange by MAFL is only made when local data is changed. If data is unchanged since the beginning (as most FL papers assume), each client exchanges averaged data only once and additional cost is incurred only in the beginning. For instance, let’s consider CIFAR-10 dataset trained with VGG-9, the same setting as the main experiment in our paper, but with $N=10^{10}$ number of clients, as suggested. Here we assume that $M_k=n_k$ just for simplicity. Then, each client exchanges whole model parameters with the global server for **every** update round, whose number is $p_m=3,491,530$. Thus, when trained for $M$ communication rounds, the total number of communication costs incurred by the transferal of model parameters is $2 \times N \times M \times p_m$ (factor of 2 multiplied for sending back and forth to the server). In contrast, the additional communication cost required for exchange of mean data under MAFL is one input data size per client, which is $p_i=3,072$ values. So, the cost of exchanging mean input data is $2 \times N \times p_i$, which is only $p_i/(M \times p_m) \approx 0.001/M$ of the cost incurred by exchanging model parameters. This ratio is significantly small that we can practically ignore additional costs incurred by exchanging mean data.
>
> As another extreme, let us consider a situation in which local data changes and is exchanged every round. In this case, communication cost incurred by exchanging mean data is multiplied by $M$. Thus, the ratio between the two costs in the previous example becomes approximately 0.001, which is still very small. So we can claim that the extra communication burden incurred by MAFL is negligible compared to the communication cost incurred by an exchange of model parameters.
>
> In general situations, model parameters are much larger than input dimension so that $p_m \gg p_i$, so the conclusion above will hold for most cases.
>
> 2.How does MAFL split data (e.g., random split)?
>
> In fact, there are no restrictions on how to group and average local data in MAFL. But in all our experiments, we performed random split for computing means of all local datasets.
>
> --How about the case where the only similar data (e.g., same class) averaged?
>
> The suggested grouping method is definitely an interesting strategy to consider. Toward this direction, we performed additional experiments on CIFAR-10 and compared this new grouping strategy against our original version, with the same $M_k=0.5n_k$ (remember that our non-iid setting assumes each client has two classes). To summarize, random split results in 81.2% accuracy while split by class results in 78.8% accuracy. The result demonstrates that splitting by class to compute mean causes a performance degradation. We can conjecture that the interpolation between more diverse labels gives the model stronger predictive power over various labels, although averaging for each class can better preserve the properties of data.
>
> We added the results in the revised version in Appendix J.
>
> 3.Consider DP version of MAFL to provide better privacy with small $M_k$.
>
> Thank you for pointing out important references and suggesting interesting direction considering differential privacy in conjunction with our mean augmentation. In our new experiments, we considered a noise level with a small $M_k$ that has privacy visually roughly similar to a maximum $M_k$. To summarize, FedMix results in 81.2% accuracy for a maximum $M_k=n_k$, while FedMix with Gaussian noise results in 81.1% accuracy for $M_k=5$ and $\sigma=0.075$ for CIFAR-10. FedMix for $M_k=5$ was 81.4%. Looking at this result alone, it seems that FedMix with noise does not perform better than that of using a maximum $M_k$. However, in addition to the small performance degradation, it is difficult to exactly quantify how much noise should be added in FedMix with noise to provide comparable privacy as the vanilla FedMix with a maximum $M_k$, since no existing theory quantifies how much privacy is provided by averaging data of differing $M_k$. While analyzing the privacy of FedMix is our longer-term goal, we believe that associating differential privacy with MAFL proposed by the reviewer could be left as future work.
>
> We included this result on DP-FedMix in the revision in Appendix I.
>
> 4.Add Global Mixup to several Tables with results.
>
> We added the ideal baselines as the reviewer proposed in our revision.

---

### Official Review · AnonReviewer3 · 2020-10-28
**An interesting idea but with some unclear statements and experiments**

**Rating:** 6
**Confidence:** 4

**Review:**

This paper studies an interesting idea that applies Mixup to Federated Learning (FL) for addressing some challenges such as non-iid data. Basically, this is an empirical paper, and the overall organization is good, easy to read. However, I have several questions.

1. The authors claimed that "FedMix approximates Global Mixup". In that case, why do not use Global Mixup directly? By comparing (4) with (3), both Global Mixup and FedMix use private image $(X_j, y_j)$, which can not violate privacy.

2. The mathematics is poorly written.
(a) What is $\ell$ in $\frac{\partial \ell}{\partial x}$? It could be $\ell(x)$, $\ell(f((1+\lambda)x_i),y_i)$, and $\ell(f((1+\lambda)x_i+\lambda x_j),y_j)$ and so on. Please write it explicitly.
(b) The last equation on Page 5 missed a $\frac{1}{|J|}$ on the left hand side since $\bar x$ or $\bar y$ means averages x_j or y_j.
(c) In proposition 1, it says "we ignore the second order term (i.e., $O(\lambda^2)$)", but why there is a $\lambda^2$ in (4)? Please check $\lambda(1-\lambda)=\lambda - \lambda^2$.
(d) In Algorithm 1 and Algorithm 2, what is k in "LocalUpdate$(k,w_t, X_g,Y_g)$" since there is no $k$ in Algorithm 2? In Algorithm 2, it seems the input is $k,w_t, X_g,Y_g$ based on "LocalUpdate$(k,w_t, X_g,Y_g)$", but what is $X, Y$ in $\ell_1$ and $\ell_2$? In addition, what is $x$ in $\ell_3$. I gauss $w$ in Algorithm 2 should be $w_t$.

3. Based on Figure 1 and Algorithms 1 and 2, $\ell_{FedMix}$ is an approximation of $\ell_{NaiveMix}$ rather than $\ell_{GlobalMixup}$, which can be easily verified by using Taylor expansion. In that case, NaiveMix and FedMix is very close when $\lambda$ is very small. However, the experimental results shows that FedMix is more closer to GlobalMixup than NaiveMix, which is not reasonable. Any explanations? Did you use the approximation of GlobalMixup as FedMix? If so, I think the presented results are not so interesting.

4. The experiments only conducted on three small data sets, but in FL, we are usually interested in big data. It would be better if the authors can provides results on big data such as ImageNet.

5. In table 8, why NaiveMix and FedMix use different $\lambda$ for CIFAR10? On the other hand, why don't you try different $\lambda$ for NaiveMix?

------------After Rebuttal------------
The authors have addressed my main concerns, and I have updated my score to 6.

---

> ### Author Response · Authors · 2020-11-13
> **Response to AnonReviewer3's Review**
>
> We thank the reviewer for a detailed review. We address the concerns below:
>
> 1.Why not use Global Mixup? Global Mixup and FedMix uses same $(X_j, y_j)$, which does not violate privacy.
>
> We believe that it is due to critical misunderstanding. Global Mixup does hurt privacy and just an **ideal** baseline. It allows the client to directly access individual data points from other clients to perform Mixup, which is not possible in the actual federated setting.
> Our FedMix, on the other hand, approximates the Global Mixup, but in a way such that it only requires the client to have access to averages of data from other clients, without violating privacy constraints. The loss of FedMix in Equation (4) is just for a single entry, but that could be expanded to a loss value for a batch that only requires averaged data from other clients (bottom of page 5 in the original paper), which is impossible for Global Mixup.
>
> 2.(a) What is $\ell$ in $\frac{\partial \ell}{\partial x}$?
>
> Since the loss function can be viewed as a function of $x$ and $y$ in the form of $\ell(f(x),y)$, $\frac{\partial \ell}{\partial x}$ is simply a derivative function of loss by input. We will clarify this in the next version.
>
> 2.(b) Divide by batch size before Sum operation >>> Thank you for the correction. We will revise accordingly.
>
> 2.(c) Why is there $\mathcal{O}(\lambda^2)$ term in $\lambda (1-\lambda)$ >>> We will correct this point.
>
> 2.(d) What is $k$ in LocalUpdate? $X,Y$ in $\ell_2$? $x$ in $\ell_3$.
>
> $k$ is the id of the client being traded, as stated in the “for” argument just above the LocalUpdate line. Algorithm 2 has $X,Y$, which is batch data selected from $X_k$ and $Y_k$, data and labels of client $k$. $x$ in $\ell_3$ is an entry from $X$, as denoted just below the line. $w$ in Algorithm 2 is $w_t$. We will revise algorithms accordingly so that all parameters are easily identifiable and understandable.
>
> 3.FedMix is an approximation of NaiveMix but performance of FedMix is closer to Global Mixup.
>
> We believe that this is also a misunderstanding.
> (1) FedMix is a valid mathematical approximation of Global Mixup (see proof for Proposition 1 of our paper in appendix). While this approximation naturally coincides with that of NaiveMix (since NaiveMix is equivalent to Global Mixup if we linearly approximate Mixup function), our only interest is that FedMix can be some approximation (even though it is rough) of Global Mixup. Therefore, it is **not** unreasonable that FedMix is close to Global Mixup compared to other algorithms not directly related to Global Mixup. In fact, it is desirable since Global Mixup is an ideal baseline.
> (2) Our results clearly demonstrate that NaiveMix < FedMix < Global Mixup in terms of accuracy, but our FedMix is much closer to our NaiveMix than ideal Global Mixup, contrary to the reviewer's concern.
>
> In fact, this question 3 is a is bit unclear to us, so it may be the case that we misunderstood your intentions. If so, please ask again with little more details so that we can better understand.
>
> 4.How about big data such as ImageNet?
>
> We agree with your concern that the scale of our experiment is below the size of big, real world federated learning applications. However, as FL is a relatively new problem to the machine learning field, the majority of papers studying FL simulated their data with small datasets such as MNIST and CIFAR-10, so it is also an issue with the majority of research papers in this area (and in our best knowledge, we did not find any FL paper with ImageNet experiments). In particular, the non-iid issue of FL we consider may occur regardless of the scale, and we expect that our mixup-based solution leveraging private data from other non-iid clients is not so sensitive to scale. For future work, we believe that our methods can be extended to the large scale datasets more relevant of real world federated learning situations.
>
> 5.Why is $\lambda$ different? Why not test different $\lambda$ for NaiveMix?
>
> We treated $\lambda$ as a hyperparameter, and picked the value that resulted in the best performance. NaiveMix was also fairly evaluated in this way. We will also append the results for NaiveMix in Table 5.

---

> > ### Comment · AnonReviewer3 · 2020-11-23
> > **Still not clear about FedMix and its loss $\ell_\text{FedMix}$.**
> >
> > Thanks for the authors' response.  It is still not clear to me about the definition of FedMix and its loss. First, if my understanding is correct, Figure 1 (d) shows that there is a averaged image $(\bar{X}_j, \bar{y}_j)$ in Global Server and it will be passed to each Client for computing loss functions.  However, I can not find $(\bar{X}_j, \bar{y}_j)$ in $\ell_\text{FedMix}$ of equation (4) from Proposition 1. Second, if $\ell_\text{FedMix}$ is an approximation of $\ell_\text{GlobalMixup}$, we can see from (4)  that FedMix will violate privacy since raw data is exchanged and directly used for computiong loss function by local data ($x_i$) and received data ($x_j$). The reason is that the last term of (4) is obtained by using both $x_i$ and $x_j$. If this is the truth, why do not use GlobalMixup directly?

---

> > > ### Author Response · Authors · 2020-11-23
> > > **Comment for Comment about $\ell_{\mathtt{FedMix}}$**
> > >
> > > Dear AnonReviewer3,
> > >
> > > We sincerely thank you for your time and effort in additional comments.
> > >
> > > We understand that your confusion comes from the fact that in Proposition 1, Equation (4), we have $\ell_{\mathtt{FedMix}}$ to use raw $x_j,y_j$. But, please note that **$\ell_{\mathtt{FedMix}}$ does not directly utilize Equation (4)**.  While this formula uses raw input, the core point of the paper is that this *approximated* formula is **linear** by $x_j$ and $y_j$. So what we can claim from this proposition is that if set of $M$ private instances from client $j$ is used to calculate the loss function, the average value of $\ell_{\mathtt{FedMix}}$ should be expressed by only using averaged $\bar{x}_j,\bar{y}_j$ and not the individual raw instances. The loss function of Global Mixup does not have this property and thus its loss function cannot be estimated with only $\bar{x}_j,\bar{y}_j$. This explanation is just below Proposition 1 (page 5) of our paper. We hope this solved your question.
> > >
> > > Please let us know if there is anything else that you want to discuss further.
> > >
> > > Thank you again for your time and please keep safe!

---

> > > > ### Comment · AnonReviewer3 · 2020-11-23
> > > > **About the implementation**
> > > >
> > > > Thanks. Now I understand. As you mentioned, $\ell_\text{FedMix}$ does not directly utilize Eq. (4), so I would like to recommend the authors to modify it in the revision. My another concern is about your implementation. Since you are using the equation after Proposition 1 as the loss of FedMix, but you missed a $\frac{1}{|J|}$ in your first version,  I am wondering if you implemented the experiments correctly?  Besides, your original Algorithms 1 and 2 are also problematic.

---

> > > > > ### Author Response · Authors · 2020-11-23
> > > > > **Comment for Comment about Formulas**
> > > > >
> > > > > > $\ell_{FedMix}$ does not directly utilize Eq. (4), so I would like to recommend the authors to modify it in the revision.
> > > > >
> > > > > - Thank you for your suggestion. We have explicitly mentioned that our loss function is based on equation (5), not equation (4), to avoid any confusion, in the most recent revision that has been uploaded.
> > > > >
> > > > > > My another concern is about your implementation. Since you are using the equation after Proposition 1 as the loss of FedMix, but you missed a $1/|J|$ in your first version, I am wondering if you implemented the experiments correctly?
> > > > >
> > > > > - In the original submission, we missed $1/|J|$ in the first line of equation (5). However, it is evident that this is merely a typo, since in the second line of (5) we **consider $1/|J|$ back**when computing the average (that is, $1/|J| \sum_j x_j = \bar{x}_j$). We revised the typo, and we have **used the second line of (5)**in our implementation.
> > > > >
> > > > > - We admit that there were some minor typos and errors in Algorithms 1 and 2 (the core of algorithms was not the problem, but rather we somehow vaguely expressed what we actually did). However, we had already corrected them in the previous revision we uploaded. We assert that our experiments were correctly conducted using the algorithms in the current version.

---

> > > > ### Author Response · Authors · 2020-11-25
> > > > **Reminder for Our Additional Comment**
> > > >
> > > > Dear AnonReviewer3,
> > > >
> > > > We think we have addressed your additional concerns in our additional comment. Since there is little time left for rebuttal, please check and let us know if there are any more problems.

---

### Author Response · Authors · 2020-11-13
**General Response**

Thank you all for helpful suggestions!

We thank all the reviewers for their thorough and helpful comments. Reflecting reviewers' suggestions, we are conducting new experiments and are revising our paper to address all concerns.

However, in the mean time, we would like to first respond to some of concerns that are trivial or caused by misunderstanding. If any of our responses to individual reviewers below is insufficient, please feel free to ask any additional questions.

---

### Author Response · Authors · 2020-11-23
**The end of the discussion phase approaching**

Dear Reviewers and Area Chair,

Could you please go over our responses and the revision since we can have interactions with you only by this Tuesday (24th)? We have responded to your comments and faithfully reflected them in the revision. We sincerely thank you for your time and efforts in reviewing our paper, and your insightful and constructive comments.

Thanks, Authors

---

### Author Response · Authors · 2020-11-23
**Summary of Changes in the Revision**

We thank all the reviewers for their thorough and insightful feedback. We revised our paper following the comments by making the following changes below. The changes are highlighted in our revision in **red** color.

**Major Updates**

* We have added an individual subsection (Section 3.4) devoted to discussion of privacy concerns. We pointed out several concerns that could be brought up by exchanging averaged data. We also suggested simple solutions that could help alleviate those issues. (R4)

* We have added estimates of additional communication cost and computation cost incurred by FedMix in Section 3.4 and Appendix G. (R1,R4)

* We revised Algorithm 2 so that it is more easily understandable and be compared to baseline algorithms (marked in **blue** color). (R3, R4).

* We added details of additional "holistic" experiment for FEMNIST dataset with different number of data per clients, as suggested by R4 in Appendix H. (R4)

* We included an experiment that compares between random split and same class split in Appendix J. (R1)

* We added a differential privacy (DP) version of FedMix on small $M_k$ in Appendix I. (R1)

* We included an experiment that tests how entries in $X_g$ could be further averaged to better ensure privacy in Appendix J. (R4)

* We added results of a newly created heterogeneous environment using Dirichlet distribution, as suggested by R1 in Appendix J. (R1)

**Minor Updates**

* We added Global Mixup results for all tables that R1 pointed out. In addition, we added a short insight about why Global Mixup is ineffective for experiment with Shakespeare dataset in Section 4.2. (R1)

* We added result of NaiveMix with varying fixed values of $\lambda$ in Appendix J. (R3)

* We fixed all minor typos and errors in text and formulas. We also added appropriate references as suggested. (all reviewers)

We claim that our paper got much stronger and clarified with this revision. We again greatly thank the reviewers for all the constructive suggestions.

---

### Decision · Program_Chairs · 2021-01-07
**Final Decision**

**Decision:**

Accept (Poster)

**Comment:**

The paper proposes to apply Mixup to Federated Learning (FL) for addressing the challenge of non-iid data. The idea is very simple, but seems to work well in empirical evaluation. Some concerns were raised regarding the communication costs and privacy. The authors rebuttal and revised draft provide reasonable answers to these concerns.

For the final version, it is suggested that the authors can address the following issues:

1) Improve the writing - especially the formulation of the proposed method

2) Provide more discussions and experiments on the communication costs.